# Variations in the poly-histidine repeat motif of HOXA1 contribute to bicuspid aortic valve in mouse and zebrafish

Gaëlle Odelin [1,10], Adèle Faucherre [2,10], Damien Marchese [3,10], Amélie Pinard[1], Hager Jaouadi[1], Solena Le Scouarnec[4], FranceGenRef Consortium*, Raphaël Chiarelli[3], Younes Achouri[5], Emilie Faure[1], Marine Herbane[1], Alexis Théron [1,6], Jean-François Avierinos[1,7], Chris Jopling[2], Gwenaëlle Collod-Béroud [1], René Rezsohazy [3] & Stéphane Zaffran [1] ✉

Bicuspid aortic valve (BAV), the most common cardiovascular malformation occurs in 0.5–1.2% of the population. Although highly heritable, few causal mutations have been identified in BAV patients. Here, we report the targeted sequencing of *HOXA1* in a cohort of BAV patients and the identification of rare indel variants in the homopolymeric histidine tract of HOXA1. In vitro analysis shows that disruption of this motif leads to a significant reduction in protein half-life and defective transcriptional activity of HOXA1. In zebrafish, targeting *hoxa1a* ortholog results in aortic valve defects. In vivo assays indicates that these variants behave as dominant negatives leading abnormal valve development. In mice, deletion of *Hoxa1* leads to BAV with a very small, rudimentary non-coronary leaflet. We also show that 17% of homozygous *Hoxa1$^{-1His}$* knock-in mice present similar phenotype. Genetic lineage tracing in *Hoxa1$^{-/-}$* mutant mice reveals an abnormal reduction of neural crest-derived cells in the valve leaflet, which is caused by a failure of early migration of these cells.

Bicuspid aortic valve (BAV) is the most common congenital heart defect (CHD) with an overall frequency of 0.5-2%[1,2] and a third of those affected develop severe complications including aortic stenosis and regurgitation, endocarditis, ascending aortic aneurysm/dissection[3,4]. BAV is defined as a spectrum of altered aortic valves presenting with two functional cusps and less than three zones of parallel apposition without fusion[5]. BAV may appear as either an isolated defect or it can be associated with other cardiac defects such as coarctation of the aorta, ventricular septal defect, and interruption of the aortic arch[6,7].

A normal tricuspid aortic valve is formed by three cusps or leaflets (right (R) and left (L) coronary and non-coronary (NC) leaflets), three sinuses at the aortic root, a commissure between each adjacent leaflets and an interleaflet triangle between each adjacent sinuses[8]. The BAV, which occurs when the aortic valve has two leaflets, has a variability in leaflet fusion phenotypes. Three major types of BAV are distinguished depending on the number of raphes (fusion), the spatial position of leaflets and the functional status of the valve[5]. The categories of BAV are type 0 (valve with no raphe), type 1 (valve with one raphe) and type 2 (valve with two raphes). The most frequent BAV types are those with one raphe positioned between the right and left coronary (R-L) leaflets (~70%) and those with a raphe between the right and non-coronary (R-NC) leaflets (~35%). A purely BAV is when there is no raphe (type 0).

The aortic valve forms in the outflow tract region of the developing heart. As the outflow tract develops, the endocardial cushions

[1]Aix Marseille Univ, INSERM, MMG, U1251, 13005 Marseille, France. [2]Institute of Functional Genomics, University of Montpellier, CNRS, INSERM, Montpellier, France. [3]Animal Molecular and Cellular Biology group, Louvain Institute of Biomolecular Science and Technology, Université catholique de Louvain, 5 (L7.07.10) place Croix du Sud, 1348 Louvain-la-Neuve, Belgium. [4]l'institut du thorax, INSERM, CNRS, UNIV Nantes, 44007 Nantes, France. [5]Transgenesis Platform, de Duve Institute, Université Catholique de Louvain, 1200 Brussels, Belgium. [6]Service de Chirurgie Cardiaque, AP-HM, Hôpital de la Timone, 13005 Marseille, France. [7]Service de Cardiologie, AP-HM, Hôpital de la Timone, 13005 Marseille, France. [10]These authors contributed equally: Gaëlle Odelin, Adèle Faucherre, Damien Marchese. *A list of authors and their affiliations appears at the end of the paper. ✉e-mail: stephane.zaffran@univ-amu.fr

become populated with mesenchymal cells, which are derived from different sources. A proportion of the cells originate from endothelial cells, which detach from one another to yield mesenchymal cells via a process named endothelial-to-mesenchymal transition (EndoMT)[9]. Neural crest cells are also responsible for populating the endocardial cushions and patterning the arterial valve leaflets[10–12]. The last source of mesenchymal cells comes from the second heart field (SHF), which has been observed during the formation of the intercalated cushion[13,14]. In the aortic valve, the right and left coronary leaflets derive from the main cushions (Neural crest/endothelial derived) while the non-coronary leaflet originates exclusively from the superior cushion (Neural crest/endothelial/SHF derived).

Genetic studies have established that BAV is a heritable trait, with about 9% of prevalence in first-degree relatives and up to 24% in larger families[15]. However, only a few mutations, particularly in the *NOTCH1*, *ROBO4* and *GATA5* genes, have been directly linked to BAV in humans[4,16–19]. More recently, mutations in the *ROBO4* gene have also been shown to predispose individuals to BAV and thoracic aortic aneurysm[20]. BAV has also been reported in a number of mouse mutants, including *Nos3*, *Nkx2-5*, *Gata5*, *Krox20*, *Robo4*, *Alk2*, and *Hoxa1*[21].

In this work, we perform targeted sequencing of *HOXA1* gene in a local cohort of 333 BAV patients and identify indel variants in the homopolymeric histidine tract of the HOXA1 protein in BAV patients. Using in vitro experiments, we show that modifying the length of the histidine repeat motif results in a decrease of protein half-life and produces a dominant negative HOXA1 variant. Furthermore, expressing dominant negative *HOXA1* indel variants in zebrafish results in defective aortic valve development. We evaluate the in vivo consequence of the reduction of histidine repeat motif by establishing knock-in mouse line harboring 10 instead of 11 Histidine. Homozygous *Hoxa1^{-1His}* mice present BAV phenotype as the *Hoxa1* knock-out mice. Using *Hoxa1^{-/-}* mice we demonstrate that this phenotype is associated with a very small, rudimentary non-coronary leaflet caused by a reduced number of mesenchymal cells into the developing intercalated cushion. These data show that reduction of Hoxa1 activity contributes to BAV phenotype.

## Results

### Identification of indel variants in the poly-histidine tract of HOXA1

In total, 333 BAV-unrelated patients were screened for genetic variability in *HOXA1*. The clinical characteristics of the study participants are summarized in Table 1. As previously reported[22], there are more men (249) presented in this cohort compared with women

(84; sex ratio = 74.77%). Also, the average age, $52.39 \pm 14.94$ years, is slightly higher than previously reported for BAV cohorts[23]. Within our cohort the most common BAV morphology was type 1 with fusion between the left and right coronary cusps (R-L type) accounting for 68.16% of BAV. Additionally, 8.5% of the cases were considered as familial based on the diagnosis of BAV and/or aortic aneurysm within a patient pedigree.

We identified several potential pathogenic variants which alter the length of the poly-histidine tract present in HOXA1. A frequent variant we identified was the p.Arg73His (c.218G>A), in a poly-histidine tract ($7.53 \times 10^{-1}$ frequency in gnomAD) which alters the reference sequence $His_{65}His_{66}His_{67}His_{68}His_{69}His_{70}His_{71}His_{72}\mathbf{Arg_{73}}His_{74}$ (WT$^G$; Table 2; Fig. 1) and results in an uninterrupted stretch of 10 histidine $His_{65}His_{66}His_{67}His_{68}His_{69}His_{70}His_{71}His_{72}\mathbf{His_{73}}His_{74}$ (hereafter referred to as WT$^A$) (Table 2; Fig. 1). We also identified patients who displayed additional mutations which further altered the stretch of poly-histidine (Table 2). The majority of length variations ($n = 22$) corresponds to the deletion of 1 histidine associated with the c.218G>A polymorphism (c.[213_215delCCA;218G>A]; named −1His). We also found 5 patients with an insertion of 1 histidine. Among these 5 patients 4 had an uninterrupted stretch (c.[213_215dupCCA;218G>A]; named +1His) and 1 had a 9His1Arg1His stretch (c.[213_215dupCCA;218G]; named +1His$^{Arg}$). Lastly, we found that 2 patients harbored a deletion of 3 histidine (1 homozygous and 1 heterozygous; c.[207_215delCCACCACCA;218G] named −3His$^{Arg}$). The clinical BAV phenotype of these patients can be found in supplementary Table 1. Interestingly, BAV type 0 is more frequently observed in patients with variants in the poly-histidine tract of HOXA1 than in the general population (20% vs. 7%; $p = 0.03$)[8]. Reported frequencies in gnomAD for these insertions and deletions are in accordance with BAV frequency in the general population (Table 3). To assess the frequency of these variants, we used a control cohort composed of 856 individuals from France (The FranceGenRef panel). Using this cohort, we found that except for the −1His variant, the +1His, +1His$^{Arg}$, and −3His$^{Arg}$ variants are more commonly represented in BAV patients than in controls (Table 2; Supplementary Table 2). Taken together, our data suggest that variants in the poly-histidine tract of HOXA1 are potentially associated with BAV.

To further examine if variants in genes associated with BAV (e.g. *NOTCH1*, *GATA4*, *GATA5*, *MUC4*, and *ROBO4*) and implicated in valvular and aortic defects were present in individuals carrying deletion or insertion in the poly-histidine tract of HOXA1, we performed exome sequencing[16,17,19,24]. After exome sequencing analysis no relevant variants were identified except for three missense variants in *NOTCH1* and *GATA5* genes. One patient with deletion of 1 histidine (−1His) carried a heterozygous *NOTCH1* variant (p.Pro649Thr). The minor allele frequency (MAF) of this variant (rs780710009) is 0.00001 in gnomAD (Supplementary Table 3). This variant is located in an EGF-like repeat domain of NOTCH1 with a conflicting interpretation of pathogenicity. Another BAV patient with aneurysm was carrying a heterozygous *NOTCH1* variant (p.Thr123Met), which is considered as a polymorphism (Supplementary Table 3), and a heterozygous variant in the *GATA5* gene (p.Leu233Pro), which has already been identified in one individual with BAV but without functional evidence[17]. Thus, exome analysis showed that variants in BAV genes were not frequently found in individuals with deletion or insertion in the poly-histidine tract of HOXA1 (2/29).

### Variations in histidine tract length affect HOXA1 half-life

To evaluate the functional consequences of these variants, we quantified protein levels in HEK293T cells transfected with either wild-type or mutated *HOXA1* cDNA constructs. Cycloheximide treatment revealed that the wild-type HOXA1 protein is very stable with a half-life of over 24 h. Since no difference was detected between WT$^A$ or WT$^G$ HOXA1 proteins, we chose to use WT$^A$ HOXA1 as the wild-type reference for all experiments. In contrast, HOXA1 proteins harboring the

## Table 1 | Baseline patient characteristics in 333 patients with bicuspid aortic valve

| Patients (n) | 333 |
| --- | --- |
| Mean age (years) | $52.39 \pm 14.94$ |
| Male (%) | 249 (74.77%) |
| Family history of BAV (%) | 30 (9.01%) |
| Type of BAV according to Sievers classification | |
| Type 0 | 44 (13.22%) |
| Type 1, R-L | 227 (68.16%) |
| Type 1, R-NC | 38 (11.42%) |
| Type 1, L-NC | 9 (2.70%) |
| Type 2 | 5 (1.50%) |
| Type 1, Not determined | 1 (0.30%) |
| Not determined | 9 (2.70%) |

*BAV* bicuspid aortic valve, *Type 0* no fusion, *Type 1, R-L* fusion of the right and left coronary (R-L) leaflets, *Type 1, R-NC* fusion of the right and non-coronary (R-NC) leaflets, *Type 1, L-NC* fusion of the left and non-coronary (R-NC) leaflets, *Type 2* fusion of the right and left coronary (R-L) leaflets and of the left and non-coronary (R-NC) leaflets.

**Table 2 | Distribution of *HOXA1* histidine stretch variations in our BAV population (*n* = 333 Index cases)**

| Variations | Impact on Histidine stretch | Short Names | Number of studied alleles | Frequencies in our BAV cohort (%) |
|---|---|---|---|---|
| c.[213_215dupCCA;218G>A] | 11His | +1His | 4 | 0.60 |
| c.[213_215dupCCA;218G] | 9His1Arg1His | +1His[Arg] | 1 | 0.15 |
| c.218G>A | 10His | WT[A] | 537 | 80.63 |
| c.218G | 8His1Arg1His | WT[G] | 99 | 14.87 |
| c.[213_215delCCA;218G>A] | 9His | −1His | 22 | 3.30 |
| c.[213_215delCCA;218G] | 7His1Arg1His | −1His[Arg] | Not found in this cohort | |
| c.[207_215delCCACCACCA;218G>A] | 7His | −3His | Not found in this cohort | |
| c.[207_215delCCACCACCA;218G] | 5His1Arg1His | −3His[Arg] | 3 | 0.45 |

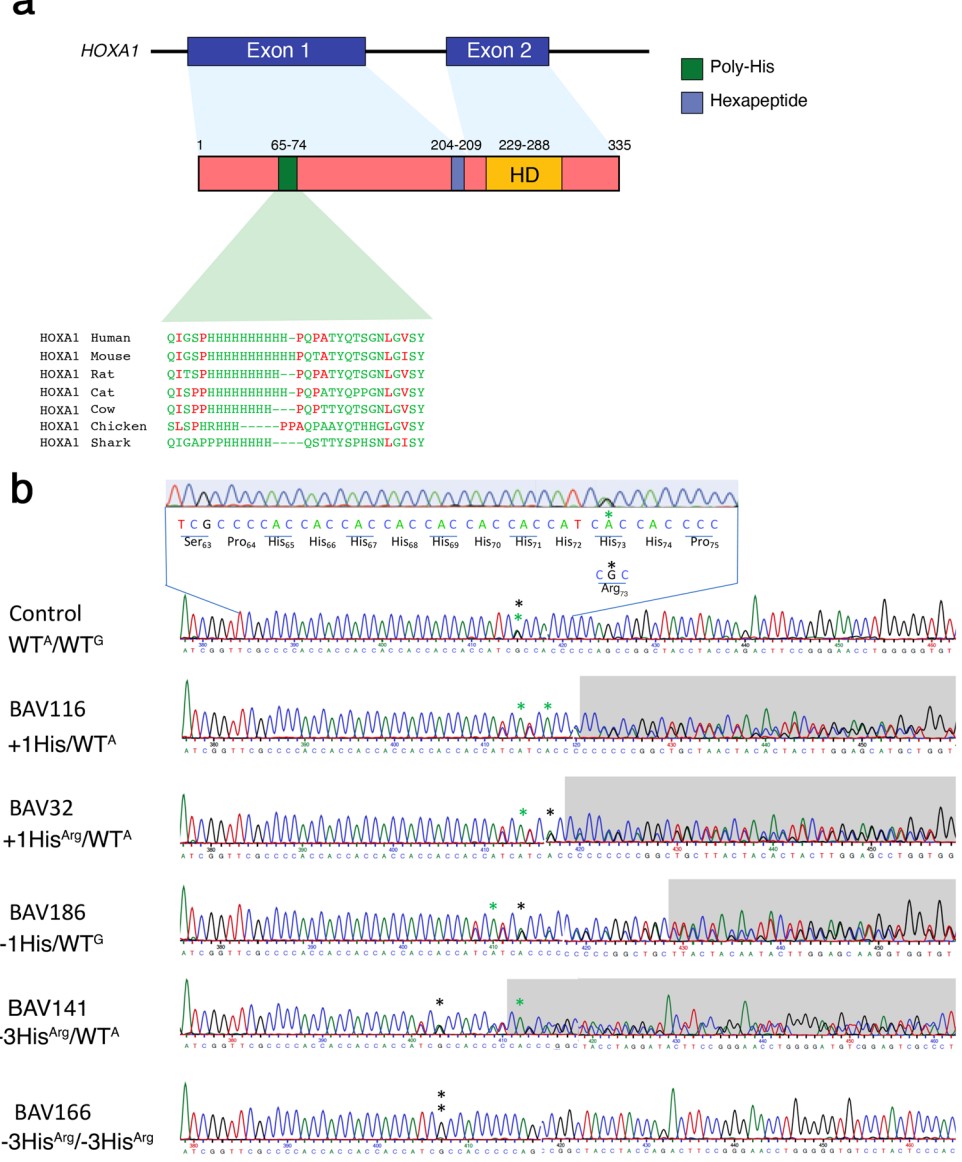

**Fig. 1 | Identification of *HOXA1* variants in index cases with BAV. a** Schematic representation of the human *HOXA1* gene and corresponding predicted protein. The HOXA1 protein contains the histidine repeat motif (green; amino acids 65–74), the hexapeptide motif involved in the interaction with PBX proteins (blue; amino acids 204–209), encoded in exon 1, and the DNA-binding homeodomain (HD; yellow; amino acids 229-288), encoded in exon 2. Alignment of the HOXA1 histidine repeat sequences from the human, mouse, rat, cat, cow, chicken and shark species is represented. The NCBI GenBank accession numbers that were utilized for the alignment are as follows: Human-P49639, Mouse-P09022, Rat-O08656, Cat-M3W8N7, Cow-E1B918, Chicken-R4GHI8, and Shark-C7B9D2. **b** Sanger sequences showing the insertion or deletion variants observed in exon 1 of *HOXA1*. The polymorphism c.218G>A (p.Arg73His) is shown with asterisks (black, c.218G; green, c.218G>A). We identified different variants in this region (c.[213_215dupCCA;218G>A] named +1His; c.[213_215dupCCA;218G] named +1His[Arg]; c.[213_215delCCA;218G>A] named −1His and c.[207_215delCCACCACCA;218 G] named −3His[Arg]) with low frequency in gnomAD (see Table 3).

**Table 3 | Reported *HOXA1* variation frequencies in the Genome Aggregation Database (gnomAD)**

| Variations | Impact on Histidine stretch | Short Names | gnomAD allele frequencies |
|---|---|---|---|
| c.[213_215dupCCA] | 11His or 9His1Arg1His | +1His or +1His[Arg] | $4.25 \times 10^{-3}$ |
| c.[213_215delCCA] | 9His or 7His1Arg1His | −1His or −1His[Arg] | $7.78 \times 10^{-4}$ |
| c.[207_215delCCACCACCA] | 7His or 5His1Arg1His | −3His or −3His[Arg] | $3.29 \times 10^{-3}$ |
| c.218 G > A | 10His | WT[A] | $7.53 \times 10^{-1}$ |

different variants altering the poly-histidine stretch displayed a markedly reduced half-life ($t_{1/2} = 5$ h versus >24 h, Fig. 2a, b). Next, we addressed whether variant HOXA1-dependent protein reduction was a result of proteasome-mediated degradation. Indeed, treating *HOXA1* transfected cells with the 20 S proteasome inhibitor MG132 (7 μM) for 20 h, dramatically increased the half-life of all the variants (Fig. 2c, d). Therefore, our results indicate that variations in the length of the homopolymeric histidine tract of HOXA1 drastically reduces protein half-life through degradation by a proteasome-dependent degradation pathway.

To investigate the impact of the histidine tract variations on the transcriptional activity of HOXA1, we used a responsive target enhancer, the *somatostatin* TSEII enhancer[25]. Based on this, we assessed the activity of wild-type and mutated HOXA1 proteins using a luciferase reporter gene controlled by the TSEII enhancer. PBX1a and PREP proteins are known to be critical cofactors of HOXA1 and to play an important role in HOXA1-mediated transcriptional activation[26,27]. HEK293T cells were co-transfected with different vectors for each of the HOXA1 proteins in conjunction with PBX1a and PREP and the TSEII-luciferase reporter plasmid (Fig. 2e, f). In comparison to wild-type HOXA1, all HOXA1 variants showed a significant decrease in transcription activity (Fig. 2e). These results indicate that variations in the length of the HOXA1 histidine motif impairs transactivation activity. Because the majority of BAV patients are heterozygous for their respective histidine variants, we next sought to determine whether these variants are associated with a dominant negative effect. To address this hypothesis, we co-transfected wild-type HOXA1 with either +1His, +1His[Arg], −1His, −1His[Arg] or −3His[Arg] HOXA1 variants. Consequently, we were able to determine that there was a significant decrease in wild-type HOXA1 transcriptional activity upon co-transfection with any of the histidine variants (Fig. 2f). Taken together these data indicate that the HOXA1 proteins with variations in the length of the histidine tract act as dominant negatives.

## Loss of *hoxa1a* results in defective valve development in zebrafish

The in vivo effects of the *HOXA1* variants were evaluated using a zebrafish (*Danio rerio*) model. Zebrafish has one *HOXA1* ortholog, *hoxa1a* (Supplementary Fig. 1)[28]. *Hoxa1a* is expressed in the midbrain and anterior hindbrain during early zebrafish development, which corresponds to a more rostral extension of expression than for *HOXA1* in mammals (Supplementary Fig. 1)[29]. Moreover, functional data suggest that a paralog, *hoxb1b*, may actually perform the same hindbrain patterning role as the mouse *Hoxa1*[28]. Therefore, we targeted both *hoxa1a* and *hoxb1b* using antisense morpholino oligonucleotides (MO) targeting the AUG start site to block translation. At 3 days post-fertilization (dpf), both *hoxa1a* and *hoxb1b* morphants displayed gross developmental defects including a heart edema (Supplementary Fig. 2). We also analyzed the developing aortic valves to determine whether *hoxa1a* or *hoxb1b* is involved in this process. To achieve this, arterial valves were imaged at 7 dpf using two-photon microscopy as previously described[30]. Wild-type larvae developed normally with two defined leaflets (Fig. 3a, d). In comparison, the valves in *hoxa1a* and *hoxb1b* morphants are highly dysmorphic and appear to be enlarged and misshaped (Fig. 3b, e). To validate that the morphant phenotype was a result of mRNA knockdown and not due to non-specific effects, we also

targeted genomic *hoxa1a* or *hoxb1b* sequences using "crispant" technology[31]. Briefly, four non-overlapping gRNAs targeting either *hoxa1a* or *hoxb1b* were injected into single cell zebrafish embryos resulting in F0 knock-outs for both genes (crispants). Consistent with our observations in morphants, aortic valve development was also defective in *hoxa1a* and *hoxb1b* crispants (Supplementary Fig. 3). We next sought to rescue the *hoxa1a* and *hoxb1b* morphants using human wild-type *HOXA1* mRNA which is not recognized by either morpholino. We found that co-injection of human *HOXA1* mRNA at 25 pg with *hoxa1a*-MO could rescue the valvular defects observed in *hoxa1a* morphants (Fig. 3c, g). Interestingly, human *HOXA1* mRNA could not rescue the *hoxb1b*-MO phenotype (Fig.3f, h; Supplementary Fig. 4). These data indicate that loss of *hoxa1a* results in defective valve development which can be reversed by human wild-type *HOXA1* mRNA.

Next, we evaluated whether the expression of the dominant negative human *HOXA1* variants in the zebrafish could also affect aortic valve development. To achieve this, we injected either the wild-type human *HOXA1* or the *HOXA1* variants at 5 pg or 25 pg of mRNA (Fig. 3i–n; Supplementary Fig. 5). Expressing any of the dominant negative variants (+1His, −1His or −3His[Arg]) also had a significant impact on aortic valve development when compared to wild-type *HOXA1* expression (Fig. 3n). Our results therefore show that expression of either +1His, −1His or −3His[Arg] HOXA1 disrupts aortic valve development in zebrafish.

## Hoxa1 contributes to aortic valve development in mice

Studies in humans found severe cardiovascular anomalies including tetralogy of Fallot, ventricular septal defect and BAV in patients carrying a homozygous truncating mutation in *HOXA1* [Bosley-Salih-Alorainy (BSAS)[32]; Athabascan Brainstem Dysgenesis Syndrome (ABDS)[33]]. Interestingly, BAV has also been reported in *Hoxa1* knock-out (*Hoxa1−/−*) mice with a limited description[34]. Strikingly, however, the expression pattern of *Hoxa1* has been reported in cell types that contribute to cardiovascular development including neural crest and second heart field (SHF) cells, but not in the developing heart per se[34–38]. Therefore, we further examined *Hoxa1*-null mice during development to better characterize the BAV phenotype. Since the leaflets of the aortic valve form by a progressive excavation of the endocardial cushions after embryonic day (E) E12.5[8], we examined *Hoxa1−/−* embryos after this stage. At E13.5, we observed the presence of three leaflets in both wild-type and *Hoxa1−/−* mice, however, the non-coronary leaflet appeared smaller in *Hoxa1−/−* ($n = 14$) compared to control ($n = 3$) embryos (Fig. 4a–d). At E15.5, one out of four *Hoxa1−/−* hearts displayed abnormal aortic valve morphology with minor non-coronary leaflet (Supplementary Fig. 6). At E18.5, we found that 27% of *Hoxa1−/−* embryos (3/11) had BAV (Supplementary Table 4), consistent with a previous report[34]. Anatomical examination of *Hoxa1−/−* aortic valves revealed a BAV with equal size between the two leaflets (Fig. 4f). Furthermore, three-dimensional (3D) reconstruction revealed a persistent small non-coronary leaflet in *Hoxa1−/−* compared to wild-type hearts (Fig. 4g, h). No other structural abnormalities were evident at the level of the other valves or the septa. Therefore, loss of *Hoxa1* function results in a BAV phenotype, which is associated with the persistence of a small non-coronary leaflet.

We next sought to evaluate the in vivo consequence of the poly-histidine tract length modifications of HOXA1 found in BAV patients by

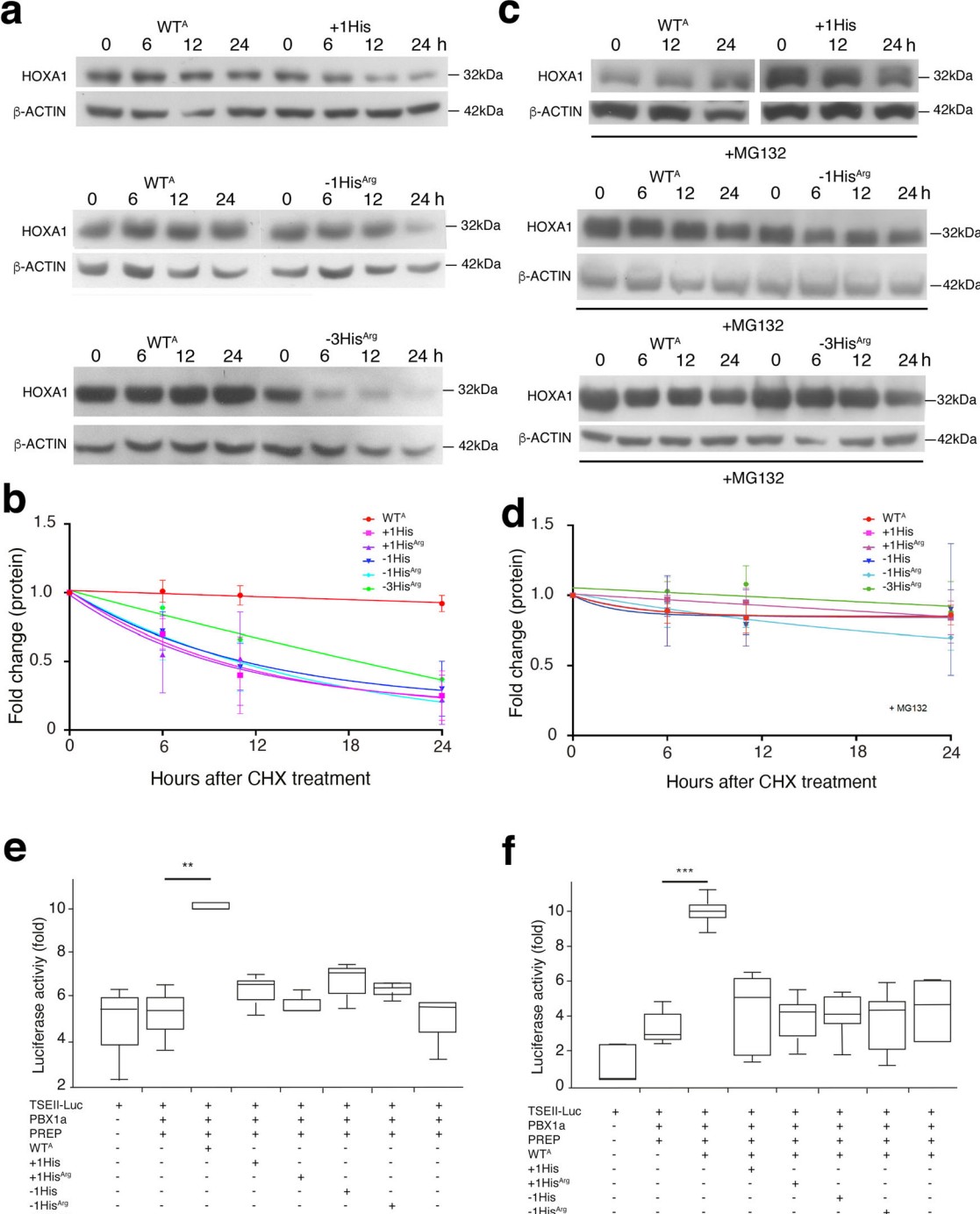

**Fig. 2 | *HOXA1* variants induce proteasome-dependent protein degradation and reduce transcriptional activity of the protein. a–d** HEK293T cells were transfected with expression vectors coding for FLAG-HOXA1 (WT[A], + 1His, +1His[Arg], −1His, −1His[Arg] and −3His[Arg]). To determine the half-life of HOXA1, cells were treated with cycloheximide (CHX; 200 μg/ml) for the indicated time. **a** Cell lysates were collected every 6 h and then subjected to immunoblot analysis with antibodies against FLAG and β-ACTIN. **b** Intensity of the HOXA1 bands observed upon cycloheximide treatment was quantified and the relative HOXA1/β-ACTIN ratios were plotted (*n* = 6; n indicates number of biologically independent experiments (cell transfected independently in different weeks). Error bars indicate standard deviation (SD). **c** Involvement of the proteasome in the HOXA1 degradation was assayed by treating cells with proteasome inhibitor (MG132; 7 μM) for 20 h prior inhibition of protein translation with CHX (*n* = 3). Cell lysates were collected every 6 or 12 h and then subjected to immunoblot analysis with antibodies against FLAG and β-ACTIN. **d** The intensity of the HOXA1 bands observed upon cycloheximide treatment was

quantified and the relative HOXA1/β-ACTIN ratios were plotted (*n* = 6). Error bars indicate standard deviation (SD). **e** Luciferase assay showing transcriptional activity of wild-type (WT[A]) and mutated hHOXA1 (+1His, +1His[Arg], −1His, −1His[Arg], and −3His[Arg]). HEK293T cells were transfected with the TSEII-luciferase reporter construct, alone (TSEII-Luc) or together with expression plasmids for PREP1, PBX1a, and FLAG-HOXA1 (WT[A], + 1His, +1His[Arg], −1His, −1His[Arg], and −3His[Arg]). Same quantity of DNA was transfected for each plasmid (*n* = 6). **p* = 0.035. **f,** In order to characterize the dominant negative effect of mutated human HOXA1 proteins, cells were transfected with the TSEII-luciferase reporter together with both wild-type and HOXA1 variant coding vectors (*n* = 6). ***p* = 0.00000012. Boxes and whiskers (min to max) show the values of lower and upper quartile. Statistical values were obtained using Pairwise *t* test with Holm *P* value adjustment method test. Statistical test used was two-sided. '*n*' represents number of biologically independent experiments (cell transfected independently in different weeks) in duplicate. Source data are provided as a Source data file.

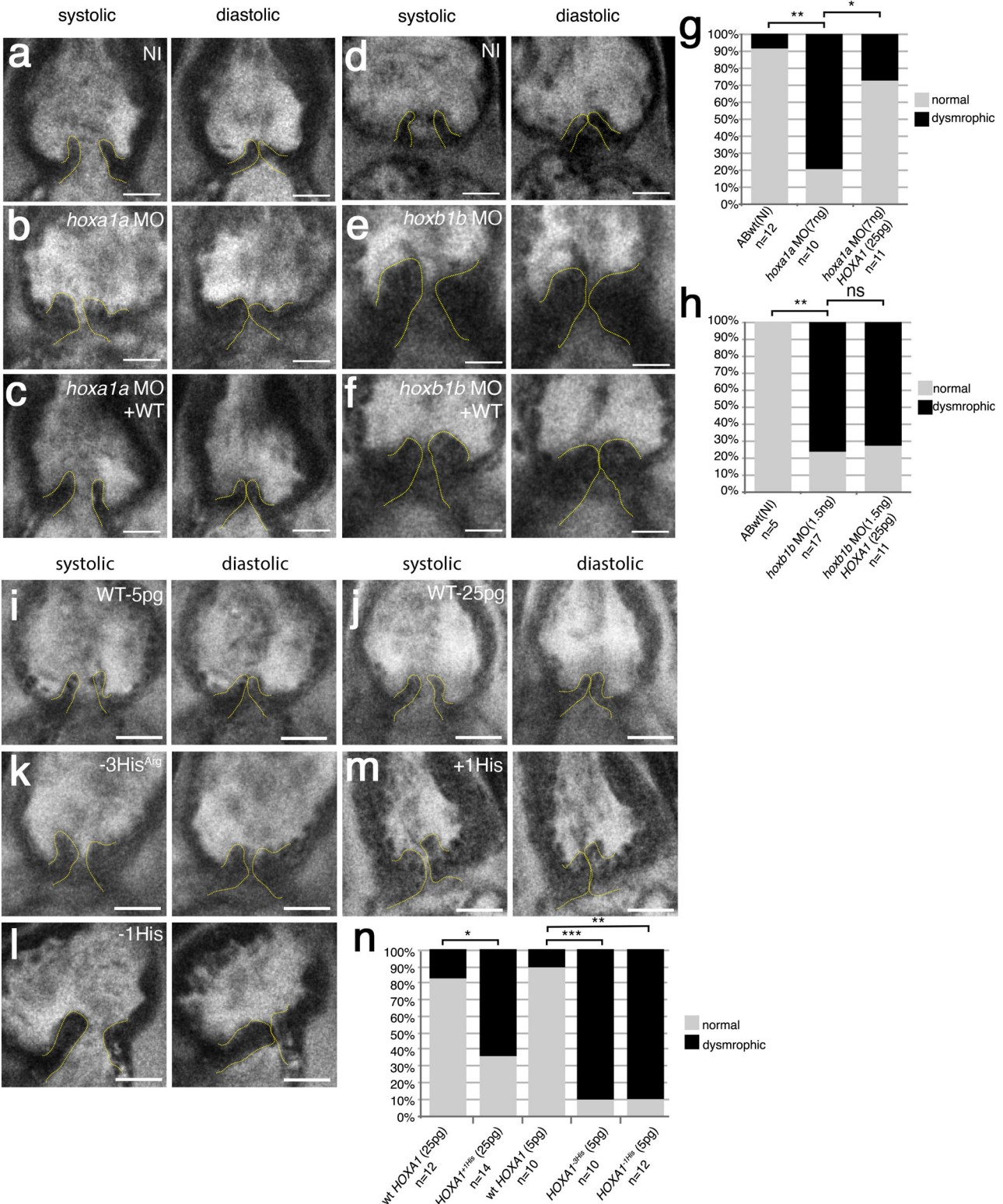

establishing CRISPR-Cas9 alleles in mouse (see "Methods"). Like *Hoxa1* knock-out, homozygous *Hoxa1*[−1His] mice exhibited a BAV phenotype with incomplete penetrance. At E18.5, 17% (4/23) of homozygous *Hoxa1*[−1His] embryos had BAV (Fig. 4i, j; Supplementary Table 4). Of note, a BAV phenotype was never observed in *Hoxa1*[+/+] (*n* = 21) or heterozygous *Hoxa1*[−1His] (*n* = 43) littermate embryos. Interestingly, 9% (5/52) of BAV was also observed in homozygous *Hoxa1*[WM-AA] knock-in embryos (Supplementary Fig. 7; Supplementary Table 4), which carry amino-acid substitutions in the hexapeptide motif of Hoxa1 that

abolish its interaction with its co-factor Pbx and its activity on known target enhancers in cellular models[25,39]. This observation suggests that Hoxa1 activity is required for aortic valve development. The incomplete penetrance of the BAV phenotype observed in *Hoxa1* knock-out and knock-in mice is in line with several mouse models of valve disease, which usually show incomplete penetrance, such as *Nkx2-5* (8.2% for BAV), *Gata5* (25% for BAV), *Krox20* (30% for BAV), and *Robo4* (17.9% in male)[12,19,40,41]. Collectively, these findings indicate that the reduction of Hoxa1 activity contributes to a BAV phenotype.

**Fig. 3 | Expression of *HOXA1* variants in vivo disrupts aortic valve development.** Two-photon images of aortic valves in 7 dpf zebrafish larvae labeled with BODIPY. **a, d** Representative image of aortic valve leaflets in a wild-type, non-injected larvae (ABwt, NI), valves are outlined with a dashed yellow line (a: *n* = 12; d: *n* = 5). **b** *hoxa1a* morphant (*n* = 10). **c** *hoxa1a* morphant co-injected with wild-type human *HOXA1* mRNA (*n* = 11). **e** *hoxb1b* morphant (*n* = 17). **f** *hoxb1b* morphant co-injected with wild-type human *HOXA1* mRNA (*n* = 11). **g** Graph showing quantification of aortic valve defects observed in 7 dpf larvae after knockdown of *hoxa1a*. Approximately 80% of *hoxa1a* morphants exhibited aortic valve defects (**p* = 0.0015; *p* = 0.03). *HOXA1* human mRNA rescued the *hoxa1a* morpholino aortic valve phenotype. **h** Graph

showing quantification of aortic valve defects observed in 7 dpf larvae after knockdown of *hoxb1b* (***p* = 0.0048). human *HOXA1* mRNA does not rescue the *hoxb1b* morpholino aortic valve phenotype. **i, j** Representative images of aortic valve leaflets after injection of WT human *HOXA1* mRNA (WT) at 5 pg (*n* = 10) or 25 pg (*n* = 12), respectively. **k** Injection of the −3His[Arg] variant of human *HOXA1* at 5 pg (*n* = 10). **l** Injection of the −1His variant of human *HOXA1* at 5 pg (*n* = 12). **m** Injection of the +1His variant of human *HOXA1* at 25 pg (*n* = 14). **n** Graph showing quantification of aortic valve defects observed in 7 dpf larvae after injection of the variants of human *HOXA1* (***p* = 0.001; **p* = 0.0037; *p* = 0.021). Statistical values were obtained using Fisher exact test. Scale bars: 20 μm. Source data are provided as a Source data file.

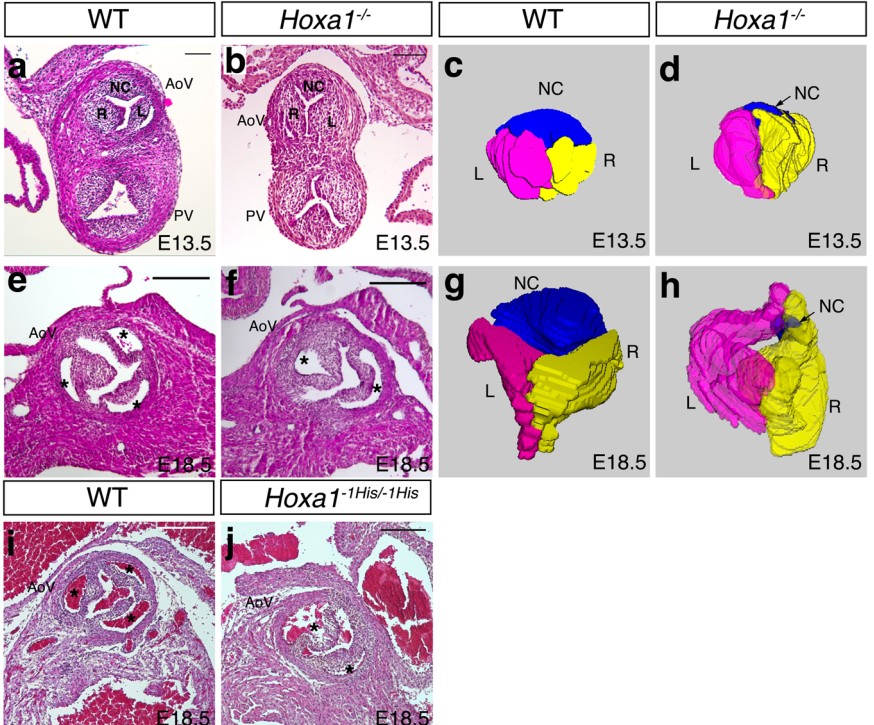

**Fig. 4 | *Hoxa1* knock-out and *Hoxa1*[-1His] knock-in mice display bicuspid aortic valve. a, b** Hematoxylin and eosin (H&E) staining images showing representative transversal section of wild-type (WT) and *Hoxa1*[−/−] outflow tract at E13.5. Representative pictures from 3 independent hearts of each genotype. **c, d** Three-dimensional (3D) reconstructions of histological images at E13.5. A slight difference between WT (**c**) and *Hoxa1*[−/−] (**d**) aortic valve (AoV) structure is observed. Indeed, the non-coronary leaflet (NC) seems smaller in *Hoxa1*[−/−] when compared to wild-type embryos. **e, f** Cross-sectional H&E images through the aortic valve of WT (**e**) and *Hoxa1*[−/−] (**f**) embryos at E18.5. Representative pictures from 3 independent hearts of each genotype. Normal valve with three leaflets (asterisks) is observed in

WT embryos (**f**), whereas bicuspid aortic valve is detected in the mutant (**f**). Asterisks indicate the aortic sinus. **g, h** 3D reconstruction of histological images at E18.5 showing three aortic valve leaflets in the WT embryo (**g**), whereas a persistent small non-coronary leaflet is seen in *Hoxa1*[−/−] embryos (**h**). **i, j** Cross-sectional H&E images through the aortic valve of WT (**i**) and *Hoxa1*[−1His/−1His] (**j**) embryos at E18.5. Representative pictures from 4 independent hearts of each genotype. Normal valve with three leaflets (asterisks) is observed in WT embryos (**i**), whereas bicuspid aortic valve is detected in the knock-in embryos (**f**). Left coronary (L; pink), right coronary (R; yellow), non-coronary (NC; blue) leaflets; PV, pulmonary valve. Scale bars: 100 μm (**a, b**); 200 μm (**e, f, I, j**).

## Decreasing of mesenchymal cells in *Hoxa1* aortic valve

The development of the arterial valves is characterized by the contribution of multiple cell types including derivatives of the endothelial, neural crest and SHF cells which are all crucial for proper valve development[11–14,42]. We first assessed whether the number of cells in the aortic valve leaflets was affected in *Hoxa1* mutant mice (Fig. 5a–c). At E13.5, we found that the number of cells in the non-coronary and left coronary leaflets was significantly decreased in *Hoxa1*[−/−] compared to control littermates (Fig. 5c; *p* < 0.001). Consistently, we observed a decrease in the relative surface of the aortic valve of *Hoxa1*[−/−] mice compared to controls (Supplementary Fig. 8; *p* < 0.001). However, cell density was unaffected in *Hoxa1*[−/−] mice suggesting that the reduction in the number of cells is not compensated by the growth of these cells (Supplementary Fig. 8).

To further analyze the contribution of neural crest, endothelial, and SHF-derived cells, we performed genetic lineage-tracing

experiments using *Wnt1-Cre*; *Tie2-Cre* and *Tnnt2-Cre* transgenic mice, respectively (Fig. 5d–l). Quantification of *Wnt1-Cre*-labeled cells showed a significant decrease of neural crest-derived cells in *Hoxa1*[−/−] aortic valve leaflets (Fig. 5d–f; *p* = 0.004). Interestingly, the non-coronary leaflet was more affected than the right and left coronary leaflets (Fig. 5f; *p* = 0.008). Similarly, quantification of *Tie2-Cre;Rosa*[tdTomato]-positive cells showed a significant decrease of endothelial derived cells in the non-coronary leaflet in *Hoxa1*[−/−] mice (Fig. 5g–i; *p* = 0.002). Genetic lineage-tracing analysis of *Tnnt2-Cre;Rosa*[tdTomato] mice revealed a decrease in the number of SHF-derived mesenchymal cells in the non-coronary leaflet in *Hoxa1*[−/−] compared to wild-type littermates (Fig. 5j–l; *p* = 0.01). On the contrary, we were unable to detect any significant difference in the number of endothelial- and SHF-derived cells in the right and left coronary leaflets in *Hoxa1*[−/−] compared to wild-type littermates (Fig.5i, l). Taken together these findings indicate that the absence of *Hoxa1* specifically affects the

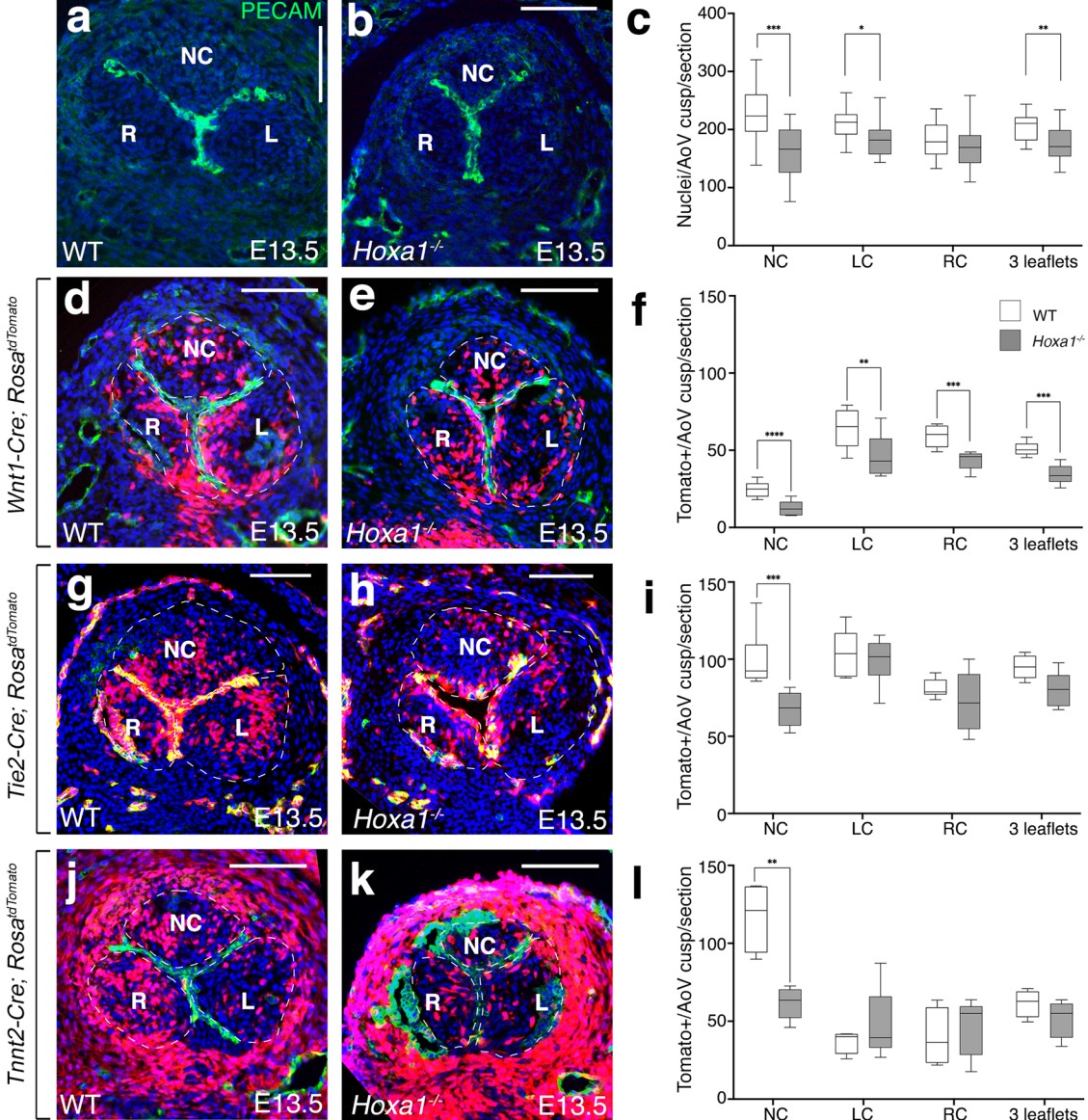

**Fig. 5 | Analysis of cell lineage contribution in *Hoxa1*-null mice. a, b** Transversal sections of E13.5 hearts immunostained with an endothelial marker (anti-Pecam antibody, green) and DAPI staining for nuclei (blue). Note the smaller non-coronary (NC) leaflet in *Hoxa1⁻/⁻* (**b**) compared with WT (**a**) embryos. **c** Total nuclei were counted in the three forming aortic valve leaflets from *Hoxa1⁻/⁻* (*n* = 20) and WT (*n* = 18) embryos at E13.5 spanning a 180 µm depth (***p* = 0.0006; ***p* = 0.0009; *p* = 0.01). **d–l** Fate-mapping of the neural crest-, endothelial- and second heart field-lineages in the aortic valves of WT (**d, g, j**), and *Hoxa1*-null (**e, h, k**) embryos at E13.5. Tomato-reporter is visualized in red and nuclei (DAPI) appear in blue.
**d, e** Immunofluorescence staining using *Wnt1-Cre;Rosa^tdTomato/+^* reporter mice shows a decrease in the number of neural crest derivatives in absence of *Hoxa1*.
**f** Quantification of neural crest-derived cells demonstrates significant decrease in *Hoxa1⁻/⁻* (*n* = 5) compared to WT (*n* = 6) littermate embryos (****p* = 0.008;****p* = 0.004; ***p* = 0.03). Area of interest is indicated by a dotted line.
**g, h** Immunofluorescence staining using *Tie2-Cre;Rosa^tdTomato/+^* reporter mice shows

a decreased contribution of endothelial lineage to the non-coronary leaflet of *Hoxa1⁻/⁻* (**h**) compared to WT (**g**) littermates. **i** Quantification confirms the decreased number of endothelial-derived cells in the non-coronary leaflet in *Hoxa1⁻/⁻* (*n* = 6) compared to WT (*n* = 6) littermates (****p* = 0.002). Area of interest is indicated by a dotted line. **j, k** Immunofluorescence staining using *Tnnt2-Cre;Rosa^tdTomato/+^* reporter mice demonstrates a decreased contribution of the second heart field lineage to the non-coronary leaflet in *Hoxa1⁻/⁻* (**k**) compared to WT (**j**) littermates. **l** Quantification confirms the reduced number of second heart cells-derived cells in the non-coronary leaflet in *Hoxa1⁻/⁻* (*n* = 5) compared to WT (*n* = 4) littermates (***p* = 0.01). Area of interest is indicated by a dotted line. Data are shown as mean ± SEM. Boxes and whiskers (min to max) show the values lower than the 25 percentile and greater than the 75 percentile. Statistical values were obtained using the Mann-Whitney test. LC: left coronary leaflet; NC: non-coronary leaflet; RC: right coronary leaflet. Scale bars: 100 µm (**a, b, d, e, g, h, j, k**). Source data are provided as a Source data file.

ability of mesenchymal cells to populate the non-coronary leaflet of the aortic valve irrespective of their origin.

### Transcriptional changes in *Hoxa1* mutant mice
To explore the molecular mechanisms leading to the BAV phenotype observed in *Hoxa1* knock-out mice, we collected tissue from the pharyngeal (branchial) regions of *Hoxa1⁻/⁻* and wild-type embryos for

RNA-seq analysis. Since previous studies revealed that neural crest and SHF cells located in the 3rd to 6th pharyngeal arches contribute to cardiovascular development, we chose to collect embryos at the early E9.5 stage when these cells are actively migrating toward the developing heart (Fig. 6a). Principal Component Analysis (PCA) indicated that samples correlate depending on their genotype (Supplementary Fig. 9). We identified 381 up-regulated genes and 205 down-regulated

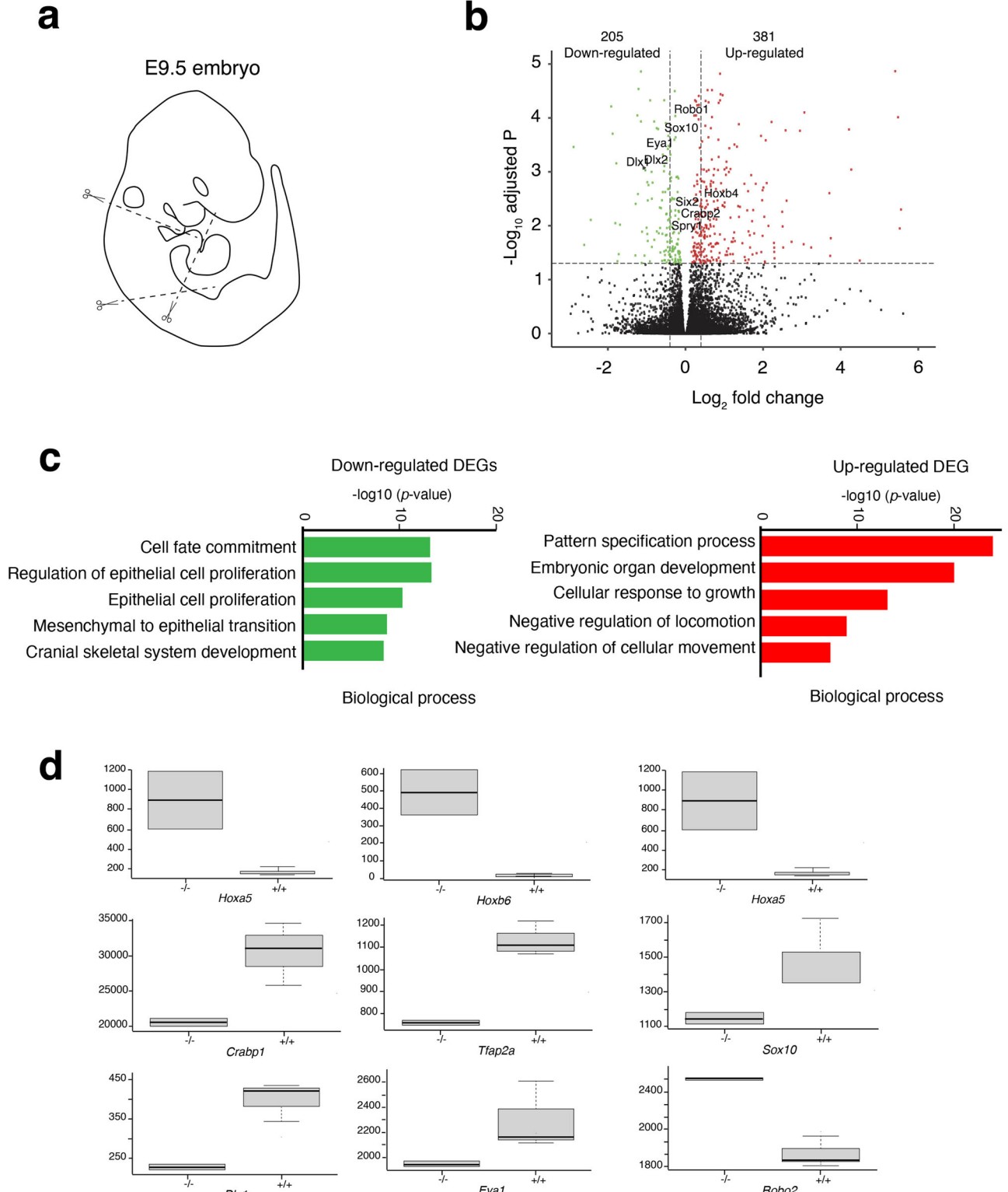

**Fig. 6 | Transcriptomic analysis identifies *Hoxa1* targets involved in migration of neural crest cells. a** Scheme showing which region was used to perform RNA-seq analysis on *Hoxa1⁻/⁻* vs WT embryos. **b** Volcano plot of fold changes and corresponding FDR (False Discovery Rate) value for each RNA after comparison of *Hoxa1⁻/⁻* and WT embryos. The red and green dots represent RNAs with fold change >2.0 and FDR < 0.05. The black dots represent RNAs with fold change <2.0 and FDR > 0.05. The statistical test used for data analysis is package R EnhancedVolcano. **c** Gene ontology (GO) analysis performed with ClusterProfiler system showing enrichment of up-regulated (red) and down-regulated (green) genes in the mutant with ranked by −log₁₀ (*p*-value). The statistical test used for data analysis is package R clusterProfiler. **d** Boxplot of FPKM (Fragments Per Kilo base of transcript per Million mapped fragments) expression value for nine selected DE genes in RNA-seq. Boxes and whiskers (min to max) show the values of lower and upper quartile. The y-axis represents the FPKM expression level, and the x-axis the genotype. Data are shown as mean ± SEM. Data are representative of three biologically independent experiments.

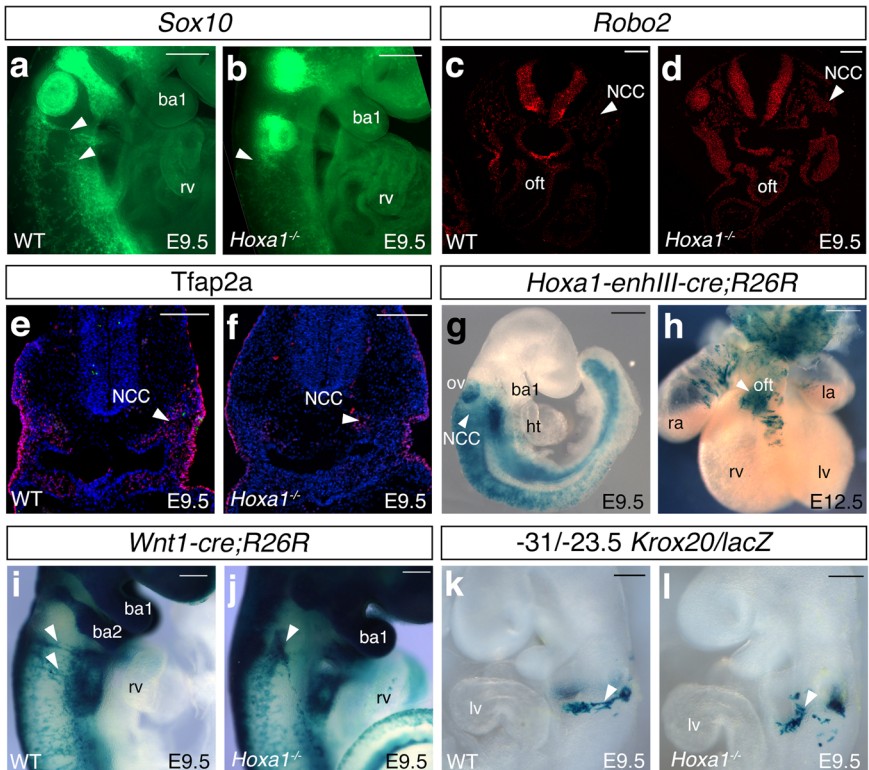

**Fig. 7 | Neural crest cell defects in *Hoxa1* null mice. a, b** RNA-FISH showing the expression of *Sox10* (green) in wild-type (WT) (**a**) and *Hoxa1⁻/⁻* (**b**) E9.5 embryos. Lateral view of *Sox10* staining revealed a decrease of expression in the migrating neural crest cells (arrowheads). **c, d**, RNA-FISH showing *Robo2* expression (green) in wild-type (**c**) and *Hoxa1⁻/⁻* (**d**) E9.5 embryos. Expression of *Robo2* is higher in *Hoxa1⁻/⁻* than wild-type littermates. **e, f** Cross-sections of wild-type (**e**) and *Hoxa1⁻/⁻* embryos (**f**) immunostained with anti-Tfap2-alpha antibody (red). Note the reduction of Tfpa2-alpha positive cells in the mutant (arrowhead). **g, h** *Hoxa1*-lineage visualized by X-gal staining of *Hoxa1-enhIII-cre;R26R-lacZ* embryos. **g**, Lateral view showing β-galactosidase activity in migrating neural crest cells (NCC; arrowhead) in transgenic embryo at E9.5. **h**, At E12.5, β-galactosidase activity is observed in the outflow tract (OFT). **i, j**, Whole-mount X-gal staining of *Wnt1-cre;R26R-lacZ* WT (**i**) and *Hoxa1⁻/⁻* (**j**) E9.5 embryos. Migration of neural crest cells (arrowhead) is disrupted in *Hoxa1⁻/⁻* embryos (**j**) compared to WT (**i**) littermate embryos. **k, l** Whole-mount X-gal staining of −31/−23.5 *Krox20/lacZ* WT (**k**) and *Hoxa1⁻/⁻* (**l**) E9.5 embryos. Migration of β-galactosidase-positive cells (arrowhead) is disrupted *Hoxa1⁻/⁻* (**l**) compared to WT (**k**) littermate embryos. ba: branchial arch; ht, heart tube; la, left atrium; lv, left ventricle; NCC, neural crest cells; ov, otic vesicle; oft, outflow tract; ra, right atrium; rv, right ventricle. Data are representative of 3 independent embryos of each genotype. Scale bars: 100 μm (**c–g**), 200 μm (**a, b, h, l**).

genes in the pharyngeal region of *Hoxa1* mutant embryos (Fig. 6b; Supplementary data 1). We selected genes significantly down-regulated and up-regulated in *Hoxa1* mutants (log2(fold change) $p > 0.2$, $p < 0.05$) to perform GO analysis and found that down-regulated genes were enriched in biological processes involved in "cell fate commitment", "epithelial cell proliferation" and "mesenchymal to epithelial transition" (Fig. 6c). In contrast, up-regulated genes were enriched in biological processes involved in "pattern specification process", "cellular response to growth" and "negative regulation of cellular movement". Interestingly, genes involved in neural crest migration were enriched in the list of down-regulated genes[43]. Thus, we found that genes involved in pre-migratory and migratory neural crest cells such as *Crabp1*, *Tfap2a* (AP-2α) and *Sox10* were down-regulated in *Hoxa1⁻/⁻* embryos (Fig. 6d; Fig. 7; Supplementary Fig. 10). Consistent with abnormal specification of neural crest cells, we found that genes normally activated in the preplacodal region lateral to the neural crest domain, such as *Eya1* and its activators *Dlx5/6* which act as lateral neural border specifiers were down-regulated (Supplementary data 1). Interestingly, *Dlx1* which is expressed in the entire mesenchyme of pharyngeal arches was also down-regulated in *Hoxa1⁻/⁻*. To further characterize the expression of genes involved in neural crest cells migration we performed RNAscope fluorescent in situ hybridization (RNA-FISH) and immunofluorescent staining. At E9.5, RNA-FISH with *Sox10* probes showed that migration of neural crest cells is largely compromised (Fig. 7a, b). Interestingly, among the up-regulated genes, we identified several posterior *Hox* genes suggesting that Hoxa1

function is necessary to establish or maintain the expression patterns of posterior *Hox* genes (Fig. 6d). In addition, we also found that *Robo1* and *Robo2*, which have been shown to affect neural crest cell migration[44,45], were up-regulated in *Hoxa1⁻/⁻* compared to controls (Figs. 6d, 7c, d). These data are consistent with previous studies showing that neural crest cell migration is affected when *Hoxa1* function is lost, a critical process for normal aortic valve formation[11].

**Abnormal neural crest migration in *hoxa1a***

To determine whether neural crest migration was also affected in zebrafish, we examined *hoxa1a* morphants and crispants in a Tg(*sox10:eGFP*) line. At 24 or 48 hpf, we did not detect any obvious difference between injected and non-injected fish (Supplementary Fig. 11). However, at 3dpf, we observed a GFP signal in the pharyngeal arch and OFT region which was absent from the morphant and crispant specimens (Supplementary Fig. 12a–c). We next sought to investigate whether overexpression of the variants could also affect *sox10* patterning. Injection of the WT or +1His variant mRNA did not seem to affect GFP expression (Supplementary Fig. 12d, e). However, both −1His or −3His^Arg induced a loss of GFP expression in this area (Supplementary Fig. 12f, g) suggesting that, similar to the mouse, *hoxa1a* could be involved in neural crest cell migration.

*Hoxa1* is strongly expressed in the posterior hindbrain where cardiac neural crest cells arise[35,38]. To determine the contribution of Hoxa1-positive cell to the formation of the outflow tract and the aortic valve, we performed genetic lineage-tracing experiment. We have

already utilized *Hoxa1-enhIII-cre* transgenic mice to demonstrate that *Hoxa1* is expressed in the SHF progenitors that contribute to the atria and part of the outflow tract[35]. At E9.5, we found X-gal positive cells in the migrating neural crest cells that invade the pharyngeal region (Fig. 7g). Consistently, we detected cells from the *Hoxa1*-lineage in the outflow tract of E12.5 embryos (Fig. 7h). Moreover, *Cre*-labeled cells were detected in the arterial valves of E13.5 and E18.5 embryos (Supplementary Fig. 13). These results confirm that *Hoxa1* is expressed in neural crest cells that contribute to arterial valve leaflet development.

### Abnormal contribution of neural crest derivatives in *Hoxa1*

Next, we investigated how loss of function of *Hoxa1* affects neural crest cells. Analysis of *Wnt1-Cre;Rosa^{lacZ}* in *Hoxa1^{-/-}* embryos revealed significant defects in cardiac neural crest cells migration (Fig. 7i, j). At E9.5, two discrete steams of *Wnt1-cre*-labeled cells were observed in control embryos (Fig. 7i), however, these streams were abnormal in *Hoxa1^{-/-}* embryos particularly into the 3rd to 6th pharyngeal arches (Fig. 7j). Immunostaining for Tfap2, a marker of migrating neural crest cells[46], confirmed abnormal numbers of cells in the 3rd pharyngeal arch (Fig. 7e, f). Recently, we found that the −31/−23.5 *Krox20/lacZ* enhancer element labels a sub-population of cardiac neural crest cells migrating from rhombomere 5-6, which contributes to the development of the aortic valve[12]. Based on these data, we generated *Hoxa1^{-/-}* mice that carry the −31/−23.5 *Krox20/lacZ* transgene to specifically examine this sub-population, and found that loss of *Hoxa1* results in aberrant migration of neural crest cells that normally migrate toward the outflow tract where they participate to aortic valve development (Fig. 7k, l). Taken together these results suggest that loss of *Hoxa1* function causes abnormal migration of neural crest cells that normally contribute to aortic valve leaflet development.

## Discussion

In this study, we identified heterozygous *HOXA1* dominant negative mutations in humans with BAV. Several studies showed that BAV is a polygenic disorder[4,24], so we cannot exclude that *HOXA1* variants act in concert with other genetic factors to contribute to the onset of the BAV phenotype. Using in vivo experiments, we demonstrated that homozygous *Hoxa1^-* knock-out and *Hoxa1^{-1His}* knock-in mice present BAV, and that targeting the zebrafish *hoxa1a* ortholog resulted in aortic valve defects that could be rescued by expressing the wild-type human *HOXA1*. The spatiotemporal expression of *Hoxa1* in the mouse, the *Hoxa1* knock-out mouse model and our RNA-seq data suggest a crucial role for *Hoxa1* in the migration of neural crest cells that contribute to the formation of the aortic valve leaflets and in particular the growth of the non-coronary leaflet.

Our study shows that variation in the length of the poly-histidine tract alters the half-life and activity of HOXA1. A comparison of HOXA1 amino-acid sequences from vertebrate species is shown in Fig. 1a and supplementary Fig. 1. This comparison displayed that in humans the poly-histidine tract in HOXA1 contains 10 His, whereas mouse Hoxa1 has a repetition of 11 His. Interestingly, we demonstrated that deletion of one histidine in the poly-histidine tract in mice leads to a BAV phenotype. Furthermore, we also observed that variations in the length of the poly-histidine tract of Hoxa1 in mice decreased the activity of the protein as shown by a luciferase reporter assay (Supplementary Fig. 14). Altogether, these finding indicate that modification of the poly-histidine motif interferes with Hoxa1 activity. Although the poly-histidine tract sequence is conserved in Shark (*Chondrichthyes*), zebrafish (*Teleost* group) lacks this domain in hoxa1a sequence. This difference can potentially be explained by a compensatory activity by other *Hox* genes during the duplication event in the lineage leading to teleosts, or by the acquisition of a neo-functionalization event specific to this species.

Single amino-acid repeats (or homopolymeric tracts) are extremely important in eukaryotic proteomes[47], and 15%-20% of human proteins contain at least one amino-acid repeat motif. Among homopolymeric tracts, histidine repeats are relatively rare in human proteins[48]. However, poly-histidine tract is more frequently associated to developmental proteins[49]. The variation of amino-acid repeats in developmental transcription factors such as PHOX2A/PHOX2B and LHX2/LHX9 has been shown to modify the transcriptional activity of these proteins similarly to HOXA1[50]. Most significantly, HOXA1 proteins with extension or reduction in the poly-histidine tract behave as dominant negatives. This is consistent with the heterozygous condition found in patients with BAV. This dominant negative effect was also validated in the zebrafish model. Indeed, injection of *HOXA1* (−1His, +1His or −3His^{Arg}) mRNA in zebrafish embryos led to abnormal arterial valve development, whereas injection of WT *HOXA1* mRNA had no effect (Fig. 3i–n). Interestingly, when we compared the proportion of aortic valve defects observed in zebrafish larvae, we found that the strongest phenotype is associated with the largest reduction of histidine (Fig. 3n).

Variation of simple amino-acid repeat tracts is often associated with human disorders[51,52]. The list of developmental and degenerative diseases that are caused by expansion of unstable repeats keeps expanding[53]. Repeat expansion diseases share number of similarities such as formation of protein aggregation and cell death. Disease-associated repeat expansions in transcription factors including HOXA13, HOXD13, and RUNX2, have been found to alter their capacity to interact with the transcription machinery to control gene expression[54]. Therefore, in addition to affecting the stability of the protein, length variation of the histidine repeat in HOXA1 could alter its interaction with other proteins involved in its transcriptional function.

Analysis of aortic valve defect observed in *Hoxa1* knock-out mice revealed an atypic morphological feature of BAV. Rather than only purely BAV with two leaflets and no raphe, our data showed that, in *Hoxa1*-null mice, a small third cushion in non-coronary position is still present, although the other two leaflets appose one another across the lumen. The presence of such structure may compromise valvular function after birth. Interestingly, the anterior leaflet of the pulmonary valve, which arises from the intercalated cushion, is not affected. Although the contribution of SHF mesenchymal cells is equivalent in both intercalated cushions, the pulmonary anterior leaflet receives fewer neural crest-derived cells than the aortic non-coronary leaflet[11,13]. This difference is likely to have a morphogenetic impact on the developing aortic and pulmonary intercalated cushions which influence their differential predisposition to pathology.

We provided evidence that neural crest cells populating the intercalated cushion are essential to promote non-coronary leaflet development. Here we showed that loss of Hoxa1 function disrupts neural crest cell migration. Previous studies have shown that *Hoxa1^{-/-}* mice have cardiovascular defects including interrupted aortic arch type B, aberrant origin of the right subclavian artery and cervical aortic arch[34,36,37]. Several studies have shown that BAV is often associated with aortopathy, aneurysmal dilation of the ascending aorta or coarctation of the aorta[55]. Since BAV and aortopathy are observed in *Hoxa1^{-/-}* mice, this suggests that a deficiency in neural crest cells migration may be a common cause of these anomalies. Indeed, neural crest cells contribute to mesenchymal cells of the aortic valve leaflets but also to smooth muscle cells of the proximal thoracic aorta[56]. Significantly, abnormal condensation of neural crest cells in outflow tract cushions during septation results in irregular sized leaflet and subsequently to BAV[10]. In addition, we have previously shown that excess of neural crest-derived cells within the aortic valve leaflets leads to BAV[12]. Therefore, we believe that reduction of neural crest cells in *Hoxa1*-null mice is the main cause of abnormal growth of the intercalated cushion and subsequent non-coronary leaflet. Unlike the main cushions, intercalated cushion formation involves three cell types including the neural crest and SHF cells[13,14]. Here, we showed that the absence of Hoxa1 specifically affects the ability of mesenchymal cells to populate the non-coronary leaflet of the aortic valve irrespective of their origin.

Several studies have shown that interaction between neural crest and SHF cells is required for normal outflow tract development[57–59]. Interaction between neural crest and SHF-derived mesenchymal cells could also be important for the growth of the intercalated cushion. However, the regulation of the interaction between these derivatives remains to be characterized. The TGF-beta/BMP signaling pathway might be involved in the interaction between the mesenchymal cells of neural crest and SHF origins. Indeed, it is important to mention that the presence of only a very small, rudimentary non-coronary leaflet has also been reported when *Alk2*, a BMP-type I receptor, is specifically deleted in post-EndoMT mesenchymal cells[60].

Our transcriptomic analysis uncovered part of the cellular and molecular mechanisms by which Hoxa1 controls neural crest cells behavior. *Hoxa1* deletion triggers downregulation of markers reminiscent of the regulatory network controlling premigration of neural crest[43]. For instance, *Tfap2a* and *Sox10*, which are part of the gene regulatory network of the pre-migratory and migratory states, are down-regulated in *Hoxa1* mutant mice. Furthermore, out of the transcriptional changes induced by the loss of *Hoxa1*, several lines of evidence support the idea that the mis-regulation of transmembrane receptor genes *Robo1*, *Robo2* and *Nrp2* is likely to play a predominant role in the decrease of neural crest cells migrating toward the pharyngeal arches 4-6. Indeed, the neural crests respond to a variety of signals that target them to a particular arch (see review[61]). Neuropilins are important not only for normal migration but also for condensation of the cardiac neural crest cells in the distal outflow tract. This later may explain the cardiovascular defects observed in *Hoxa1* mutant mice. Interestingly, the Slit/Robo signaling seems to be an attractant for arch 4 colonizing crest cells and repellant in trunk crest. We found that *Robo1* and *Robo2* are up-regulated following the loss of *Hoxa1*, which may contribute to the disruption of normal neural crest cells migration in this environment.

In conclusion, we identified indel variants in the poly-histidine tract of HOXA1 in humans with BAV and showed that length variation of this poly-histidine motif modifies the half-life and the activity of HOXA1 protein. In addition, using the zebrafish model we found that these indel variants act as dominant negatives resulting in defective aortic valve development. We showed that deletion of *Hoxa1* in the mouse results in BAV phenotype associated with abnormal growth of the intercalated cushion and subsequent non-coronary leaflet. Importantly, we detected BAV in *Hoxa1⁻¹ᴴⁱˢ* knock-in mice indicating that modification of the poly-histidine tract disturbs the activity and function of Hoxa1 protein. Since BAV identified in *Hoxa1* mutant is associated with the persistence of a small non-coronary leaflet, it would be important to use a more detailed classification describing subtle variations in valve morphology in humans. A recent international BAV nomenclature and classification consensus was proposed for universal used by clinicians, geneticists, and researchers. This simple but comprehensive nomenclature and classification system is based on imaging and proposes 3 BAV types: the fused BAV, the 2-sinus BAV and the partial fusion BAV, each with a specific phenotype[62]. This classification seems more accurate in the description of the valvular phenotype observed in humans. However, using this classification system a reduction of the non-coronary leaflet would be linked to the 2-sinus BAV phenotype (previously named BAV type 0). Our study confirms that the BAV phenotype is much more complex than what we can see and a classification that could recognize all types of BAV would be useful to better identify the genetic cause of this pathology.

## Limitations
There may be some possible limitations in this study. The first is the limited access to detailed clinical data for the majority patients. The second limitation concerns family segregation. Indeed, familial co-segregation was not possible in our study, which does not allow us to confirm a link between HOXA1 and BAV mutations in humans.

## Methods
### Patients
This study was performed according to the principles of the Declaration of Helsinki and to the ethical standards of the first author's institutional review board. The patients provided their written informed consent to participate in this study (approved by the Marseille ethic committee n°13.061). A group of 333 BAV index cases (249 males, 84 females) were prospectively recruited from La Timone Hospital, Marseille in strict compliance with all relevant ethical regulations. Patients underwent clinical evaluation that included family history, physical examination and 2D-echocardiography. Patients with another clinical disorders such as autism or syndromic clinical presentations were excluded. Genomic DNA was extracted from blood samples according to standard techniques (cat: 51206; Qiagen).

### Genetic analysis
Genomic DNA was extracted from peripheral blood leukocytes using the FlexiGene DNA kit (Qiagen) following the manufacturer's protocol. DNA purity and quantity were assessed with Nanodrop spectrophotometer (Thermo Scientific).

The two exons and intronic *HOXA1* regions were sequenced bidirectionally to search for sequence variations. Primers were designed according to *HOXA1* genomic reference sequence NM_005522. PCR products were sequenced using BigDye Terminator Cycle Sequencing kit (Applied Biosystems) and run-on automatic sequencer ABI 3130XL (Applied Biosystems). Patient sequences were aligned to the *HOXA1* reference sequence using Seqscape software V5.2 (Applied Biosystems) and validated using Sequencher V5.4.6 (Gene Codes Corporation).

Variant pathogenicity was estimated using UMD-Predictor (http://umd-predictor.eu)[63] and Human Splicing Finder[64]. Allele frequencies for each variant were obtained from the Genome Aggregation Database (gnomAD)[65], which provides high-quality genetic data on 141,456 individuals (125,748 exomes and 15,708 whole genomes).

### Detection of variants in the Histidine stretch in controls
The FranceGenRef panel comprises 856 individuals from France. Whole-genome sequencing BAM files were mapped according to the Broad Institute's Best Practices. We then used a custom java tool to genotype the interval. This tool scans the BAM files in the interval and builds the DNA sequences from the CIGAR string of each read. Each DNA sequence is then reverse complemented and translated to get the amino-acid sequence. The code for this tool is available at: https://github.com/lindenb/ScanHoxa1/ (https://doi.org/10.5281/zenodo.7625919).

### Whole exome sequencing
Genomic DNA was extracted by standard techniques. Exome sequencing was performed using the Twist Comprehensive exome (37 Mb) kit according to the manufacturer's protocol (Twist Bioscience). Paired-end 75-bp reads from the DNA libraries were sequenced using Illumina NextSeq 500 platform (Illumina). Raw fastQ files were aligned to the hg19 reference human genome (University of California Santa Cruz, UCSC) using the Maximum Exact Matches algorithm in Burrows-Wheeler Aligner (BWA) software. Alignment quality was evaluated using Qualimap 2.2.2.

Variant calling and annotation were performed using GATK and ANNOVAR best practices, respectively. Variant annotation and exome analysis were performed with VarAFT software, version 2.17 (http://varaft.eu).

### Mouse strains
Animal experiments were approved by the "comité d'éthique pour l'expérimentation animale" (agreement number C13 013 08, Marseille ethical committee, Protocol N°32-08102012) and by the "Animal

Experimentation Ethics Committee of the UCLouvain" (agreement number LA1220028, project code 202801). All animal experiments conformed to the guidelines of Directive 2010/63/EU of the European Parliament on the protection of animals used for scientific purposes. Mice were housed at 24 °C, 12 h light/dark. The null allele of *Hoxa1^neo* (*Hoxa1^tm1Ipc*, MGI:1857504, hereafter referred to as *Hoxa1*) has been described previously[66]. The knock-in allele *Hoxa1^WM-AA* (*Hoxa1^tm1Rez*; MGI:3056069) has been described previously[39]. All genotypes were observed at the expected Mendelian ratios. *Hoxa1-enhIII-Cre, Wnt1-Cre (Tg(Wnt1-GAL4)11Rth;* MGI:3524966*), Tie2-Cre (Tg(Tek-cre)12Flv,* MGI:2136412), *Tnnt2-Cre (Tg(Tnnt2-cre)5Blh;* MGI:2679081), *Rosa^LacZ (Gt(ROSA)26Sor;* MGI:1890203) and *Rosa^tdTomato (Gt(ROSA) 26Sor^tm9(CAG-tdTomato)Hze*; MGI:3809523) transgenic lines have been previously described[35,67–69]. CD1 and C57BL/6J mice were ordered from Charles River France. All animals, except *Wnt1-Cre* mice, were backcrossed on a C57BL/6J background. 8 to 10-week-old female mice were crossed with 10 to 30-week-old male mice and embryos were harvested from pregnant females sacrificed by $CO_2$ at the right stage determined by the morning of the vaginal plug as embryonic day (E) 0.5.

## Generation of *Hoxa1* knock-in mice line by CRISPR-Cas9

Cloning-free CRISPR-Cas9 system[70] was used to modify the polyhistidine tract in the mouse *Hoxa1* gene by homologous direct repair mechanism. CRISPRdirect software (http://crispr.dbcls.jp/) was used for designing the guide RNA[71]. We used 2-part guide RNA (crRNA + tracrRNA) to target the mouse *Hoxa1* gene. crRNA corresponding to the target sequence (20 nucleotides) with a tail (16 nucleotides) and a tracrRNA (67 nucleotides) containing the hairpin recognized by Cas9 nuclease with an extremity complementary to the crRNA tail. Single-strand oligonucleotide containing an inserted histidine codon (CAT) was ordered as Ultramer DNA (Hoxa1^His) and used as DNA donor template for homologous direct repair mechanism. Sequences of guide RNA and single-strand DNA oligonucleotide are shown below. Annealed crRNA/tracrRNA (2.4 pmol/ μl), recombinant Cas9 nuclease V3 (200 ng/μl) and single-strand DNA donor (10 ng/μl) were mixed in IDTE buffer and injected into the pronucleus of B6D2F2 mouse zygotes. Injected zygotes were incubated in KSOM medium at 37 °C for at least 2 h before being transferred into oviducts of CD1 pseudo-pregnant female mice. crRNA, tracrRNA (1072534), recombinant Cas9 nuclease V3 (1081059) and IDTE (11-01-02-02) were ordered from Integrated DNA Technologies (Leuven, Belgium).

Hoxa1-crRNA:

5′ CCACCATCACCACCCCCAGA 3′

Hoxa1^His Ultramer:

5′GGCAGGGGGGTGCAGATCAGCTCGCCCCACCACCACCACCAC CACCACCACCATCACCACCATCCCCAGACGGCTACTTACCAGACTT CTGGAAACCTTGGGATTTCTTATTCCCACTCGAGT 3′

## Plasmid construction

Plasmids pCS2-Prep1, pCMV-PBX1a, TSEII-Luc, pGL4.74 [from Promega] have been described elsewhere[25]. Plasmids coding for wild-type HOXA1 (human HOXA1) and hHOXA1 mutants −1His, −1His^Arg and −3His^Arg were generated with the Gateway® cloning system. Gateway® entry vectors were obtained using the following primers for PCR delineating the *hHOXA1* sequence: 5′-GGGGACAACTTTGTACAAAAAAGTTGGCATGGACAATGCAAGAAT GAACTCC-3′ and 5′-GGGGACAACTTTGTACAAGAAAGTTGGGTAGT GGGAGGTAGTCAGAGTGTC-3′.

For plasmids coding for the hHOXA1 mutants +1His and +1His^Arg, mutagenesis was carried out by NEBaseChanger using the following mutagenic primers forward 5′- CACCATCACCACCCCCAGCCG −3′, reverse 5′-GTGGTGGTGGTGGTGGGTG-3′

For plasmids coding for the hHOXA1 variants −1His, −1His^Arg and −3His^Arg, mutagenesis was carried out by overlapping triple PCRs using the above primers as well as the following mutagenic primers: +1His forward 5′-CAGATCGGTTCGCCCCACCACCACCACCACCACCATCAC-CACCCCCAGCCGGCTACC-3′, reverse 5′-GGTAGCCGGCTGGGGGTGG TGATGGTGGTGGTGGTGGTGGTGGGGGCGAACCGATCTG-3′; −1His, forward 5′-CAGATCGGTTCGCCCCACCACCACCACCACCACCATCAC-CACCCCCAGCCGGCTACC-3′, reverse 5′-GGTAGCCGGCTGGGGGTGG TGATGGTGGTGGTGGTGGTGGTGGGGGCGAACCGATCTG-3′; −3His^Arg forward 5′-CAGATCGGTTCGCCCCACCACCACCACCATCACCACCCC-CAGCCGGCTACC-3′, reverse 5′-GGTAGCCGGCTGGGGGTGGTGATGG TGGTGGTGGGGGCGAACCGATCTG-3′. BP-clonase® reactions were achieved using pDONR223 vector and amplified cDNA to generate Gateway® entry pEnt plasmids. These pEnt plasmids were involved in LR-clonase® reactions with pDest-FLAG N-ter destination vector [4] to generate pExp-FLAG-hHOXA1 expression vectors for the WT and mutant hHOXA1 sequences.

## Cell culture

HEK293T cell line (cat: CRL-3216) was purchased from ATCC. HEK293T cells were cultured in DMEM medium (cat: 61965-059, Gibco,) supplemented with 10% fetal bovine serum (cat: 10270-106, Invitrogen) and 100 U/ml penicillin-streptomycin (cat: 15140-122, Gibco), 1 mM sodium pyruvate (cat: 11360-070, Gibco). All cells were cultured at at 37 °C under 5% of $CO_2$.

## Luciferase reporter assays

For Luciferase reporter assays, HEK293T cells (cat: CRL-3216, ATCC) were seeded in 96-well plates ($1.4 \times 10^4$ cells per well) and transfected after 24 h using JetPrime transfection reagent (cat: 114-07, Polyplustransfection), with 65 ng of target luciferase reporter plasmid (TSEII-Luc), 25 ng of each FLAG-hHOXA1 expression plasmids (WT, + 1His, +1His^Arg, −1His, or −3His^Arg), 12.5 ng of PREP and PBX expression plasmids each, and 5 ng of constitutive reporter plasmid pGL4.74. Empty vector was added when needed so that each transfection included a total amount of 120 ng of DNA. Forty-Eight hours after transfection, cell lysis and enzymatic activity analysis were performed with Dual-Glo® Luciferase Assay System (cat: E2920, Promega) following manufacturer's instructions. For each transfection, the constitutively active pGL4.74 reporter was used as an internal standard for target reporter activity normalization. The relative luciferase activity was established as the ratio between target and constitutive luciferase activities.

## Protein half-life analysis

HEK293T cells (cat: CRL-3216, ATCC) were plated on 6-well plates at 700,000 cells per well. The cells were transfected with 200 ng of specific pExp plasmids coding for FLAG-tagged WT or mutant HOXA1 proteins using JetPrime transfection reagent (cat: 114-07, Polyplustransfection). For translation inhibition, 24 h after transfection, cells were treated with 200 μg/ml of cycloheximide (cat: 01810, Sigma-Aldrich). For both translation and proteasome inhibition, cells were pre-treated 20 h after transfection with 7 μM of MG132 (cat: 474790, Merck Milipore) and 24 h after transfection were treated with 200 μg/ ml of cycloheximide and with 7 μM of MG132. HEK293T cells were lysed with IPLS buffer (20 mM TrisHCl pH 7.5, 120 mM NaCl, 0.5 mM EDTA, 0.5% NP40, 10% glycerol) supplemented with Complete™ protease inhibitor (cat: 11873580001, Roche) for 20 min under agitation on ice. Cell lysates were centrifuged 5 min at 16,000 g at 4 °C. 50 μl of each lysate was mixed to Laemmli 6×, boiled 5 min at 95 °C and run on a 10% SDS-PAGE. Western blotting was used to visualize FLAG-tagged and β-ACTIN proteins with anti-Flag (1:1,000; clone F2; cat: F1804; Sigma-Aldrich), secondary anti-mouse-IgGκ BP-HRP (1:1,000; cat: sc-516102; Santacruz), and Anti-β-Actin-Peroxidase antibody (1:5,000;

clone AC-15; cat: A3854; Sigma-Aldrich). ImageJ software was used to estimate protein abundance.

## Zebrafish studies

AB wild-type (ABwt) zebrafish were maintained under standardized conditions and experiments were conducted in accordance with the European Communities council directive 2010/63. The Tg(*sox10:eGFP*) line was a gift from Drs F. Djouad and D. Sapède (IRMB, Montpellier). GFP was imaged with a confocal microscope Leica SP8 resolution 1024×512, 20X oil objective.

Morpholino oligonucleotides (MOs) were obtained from Gene Tools (Philomath, OR, USA) and injected into one-cell stage embryos. The sequence of the injected MOs are the following:

*hoxa1a* MO: 5′-CTAAGAATGTGCTCATTGTGTGTCATC-3′, 7.5 ng

*hoxb1b* MO: 5′-ATTGCTGTGTCCTGTTTTACCCATG-3′, 1 to 2 ng

Rescue experiments were performed by co-injecting *hoxa1a* MO (7.5 ng) or *hoxb1b* MO (1.5 ng) with 25 pg of human wild-type *HOXA1* (WT) mRNA. Injections of human *HOXA1* mRNA alone were performed with either 5 pg (WT, −1His, −3His$^{Arg}$) or 25 pg (WT, + 1His) of in vitro transcribed 5′ capped sense RNAs synthesis using the mMessage mMachine kit (Ambion).

Crispants were obtained by co-injecting four non-overlapping gRNAs targeting either *hoxa1a* or *hoxb1b* with Cas9 protein into single cell zebrafish embryos.

All gRNAs were prepared as described[31].

*hoxa1a* target sequences:

1-TTGAAAGTCCGTGATCACCG; 2-GGTAATTGACAAAGTCCCGG; 3-GTAAACTAACGGGCCACACT; 4-CGTACTGGAACTGGCCATGG

*hoxb1b* target sequences:

1-TCGGCACCGCACGAAACTCA; 2-CTGAACGAACCATAGCCGTG; 3-ATTGGCTCCCATATTCACGA; 4-TGTCCATAGTCCGAATGAGG

For aortic valve imaging, larvae were incubated one day prior imaging in 0.2 µM BODIPY-FL Ceramide (cat: D3521, Invitrogen) in Embryo medium + PTU (0.003% 1-phenyl-2-thiourea). 7 dpf larvae were then anesthetized with Tricaine (0.16 g/L) and mounted in low melting agarose. Imaging was performed with a Zeiss LSM710 two-photon microscope coupled to a Ti:sapphire laser (Spectra-Physics, Santa Clara, CA, USA) and a water immersion 25X objective.

## Histological, immunohistochemistry and X-gal staining

Standard histological procedures were used[72]. Heart tissues from *Hoxa1*$^{-/-}$ and littermate controls were paraffin-embedded and cut at 8 µm per tissue sections. Sections were stained with Hematoxyline & Eosine (cat: HHS32 and HT110232, Sigma-Aldrich) according to the manufacturer's instruction. CD31 (PECAM) (1:100, cat: 553370, Pharmingen) and anti-AP2-alpha (1:50, cat: 5E4, DSHB, immunostaining) immunohistochemistry were performed using 4% paraformaldehyde fixed tissue. Donkey anti-Rat Alexa 488 (1/500; cat: A21208, ThermoFisher Scientific) and donkey anti-Mouse Alexa 555 (1/500; cat: A31570, ThermoFisher Scientific) were used as secondary antibodies. X-gal staining was performed on 12 µm thick frozen sections of hearts from E13.5 and E18.5 as described previously[72]. For each experiment, a minimum of 3 embryos of each genotype was scored. Sections were examined using a DM5000 Leica microscope with 20X objective and photographed with a digital camera (LAS software, Leica).

## RNA-FISH

RNA-FISH was performed according to the protocol of the RNAscope Multiplex Fluorescent v2 Assay (cat: 323110, Acdbio), which detects single mRNA molecules. In briefly, E9.5 embryos were fixed for 20–30 h in 4% paraformaldehyde and then dehydrated in methanol. The following probes were used: mm-*Sox10*-C2 (cat: 43591-C2, Acdbio), mm-*Robo2*-C1 (cat: 475961, Acdbio). Whole-mount and sections

RNA-FISH were performed as previously described[73]. Embryos were imaged using an AxioZoom.V16 microscope (Zeiss) with ×40 objective and photographed with an Axiocam digital camera (Zen 2011, Zeiss). Sections were examined using an Apotome (Zeiss), 10X objective and photographed with an Axiocam digital camera (Zen 2011, Zeiss).

## RNA-seq

Total RNA was isolated from the pharyngeal region and sorted cells with NucleoSpin RNA XS (cat: 740902.50, Macherey-Nagel) following the protocol of the manufacturer. RNA-seq libraries were created on RNA samples (500 ng) with RNA integrity number (RIN) > 7 using 'kapa mRNA HyperPrep Kit' (cat: KR1352–v4.17, Roche). Final cDNA libraries were checked for quality and quantified using 2100 Bioanalyzer (Agilent Technologies). The libraries were loaded in the flow cell at 1.8 pM concentration and clusters were generated in the Cbot and sequenced in the Illumina NextSeq 500 as 75 bp pair-end reads following Illumina's instructions. Reads were mapped onto the GRCh38 assembly of the Mus musculus genome using STAR v2.5.3a. Aligned reads in BAM format were annotated against the protein-coding mRNA regions. Stringtie v.1.3.1c platform was utilized to visualize the annotated/mapped read, quantify data (using corrected log2 transformed Reads Per Million Values of nonstrand specific, unmerged isoform), and perform percentile normalization. An intensity difference analysis method was used to identify differentially expressed genes (DEGs) on the normalized quantification data (DESeq2 (v1.18.1).).

## Three-dimensional (3D) reconstructions

Fiji software (ImageJ 1.53q) was used to make the 3D reconstructions presented in our manuscript. At E13.5, E15.5 and E18.5, images of 20-30 8 µm paraffin sections were manually aligned and taken to generate 3D reconstruction.

## Statistical analysis

All parametric data are expressed as mean ± SEM. Statistic was determined using the Student's *t* or Fisher's exact test to compare variances. For non-parametric data, the Mann–Whitney test was used to calculate significant between the medians. A *p* value of less than 0.05 was considered significant.

## Reporting summary

Further information on research design is available in the Nature Portfolio Reporting Summary linked to this article.

## Data availability

The authors declare that all data supporting the findings of this study are available within the article and its supplementary information files or from the corresponding author upon request. The raw data for the transcriptomic data have been deposited in the Gene Expression Omnibus database from NCBI under accession code: GSE224217. The source data underlying Figs. 2a–f, 3g, h, n, 5c, f, i, l, S3 and S8 are provided as a Source Data file. Source data are provided with this paper.

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

## Acknowledgements

The authors thank Pr. Pierre Chambon for his authorization to use the *Hoxa1^neo* mouse strain. We thank N. Eudes for her technical help during HOXA1 sequencing. We are grateful to the GBiM MMG Platform and particularly to Nicolas Lenfant to his help in the analysis of the RNA-seq data. We thank all staff members of MMG animal core facility for their help in the breeding. A.F. and C.J. acknowledge the Montpellier MRI and IPAM facilities, members of the France-BioImaging national infrastructure supported by the French National Research Agency (ANR-10-INBS-04, «Investments for the future»). The FranceGenRef Consortium was supported by LABEX GENMED funded as part of "Investissement d'avenir" program managed by Agence Nationale pour la Recherche (grant number ANR-10-LABX-0013). This work was supported by the "Association Française contre les Myopathies" [NMH-Decrypt Project], the "Fondation pour la Recherche Médicale" [DPC20111123002], the "Institut National de la Santé et de la Recherche Médicale" and "la Fondation Leducq" to S.Z. This work was supported by the "Fonds de la recherche Scientifique FNRS" [Crédit de recherche (CDR) J.0157.21] and the "Fonds spéciaux de recherche" (FSR, UCLouvain) to R.R. A.P. received PhD fellowships from the "Association Française du syndrome de Marfan et apparentés". A.F. and C.J. are members of the Laboratory of Excellence Ion Channel Science and Therapeutics supported by a grant from the ANR. Work in the C.J lab is supported by a grant from the "la Fondation Leducq".

## Author contributions

G.O., A.F., D.M., R.R., and S.Z. conceived the study, interpreted the data, and wrote the manuscript. G.O., A.F., and D.M. designed and performed most of the experiments using mice, zebrafish, and molecular assays, immunostaining, and analysis data. A.P., H.J., S.L.S., and G.C.B. performed bioinformatic analyses. R.C., and Y.A. generated the CRISPR-Cas9 knock-in mouse line. E.F., and M.H. performed some molecular assays and immunostaining. A.T., and J.F.A. diagnosed and recruited patients with BAV. C.J., R.R., and S.Z. conceived and supervised the study and provides financial support. All authors read and approved the final manuscript.

## Competing interests

The authors declare no competing interests.

## Additional information

## FranceGenRef Consortium

Jean-François Deleuze[8], Emmanuelle Génin[9], Pierre Lindenbaum[4], Richard Redon[4] & Jean-Jacques Schott[4]

[8]Centre National de Recherche en Génomique Humaine, CEA, Fondation Jean Dausset-CEPH, Evry, France. [9]Inserm, Univ Brest, EFS, UMR 1078, GGB, F-29200 Brest, France.

