## [Peer Review File · Nature Communications]

Variations in the poly-histidine repeat motif of HOXA1 contribute to bicuspid aortic valve in mouse and zebrafishReviewers' comments:

Reviewer #1 (Remarks to the Author):

Significance:

The authors present data in humans, zebrafish, and mouse indicating that insertions or deletions in the poly-his tract of HOXA1 predispose to BAV. HOXA1 is essential for vascular development in humans, and this study adds further evidence for a role in human aortic valve development. The paper is generally well written and the figures are clearly presented. As such, this study provides important biological and clinical insights into the functions of Hoxa1/HOXA1 for aortic valve development.

Main Critiques:

As the authors point out, insertions or deletions in the HOXA1 poly-his tract are relatively common in humans. In the early 2000s, for example, variants in this region were investigated for association with autism. This raises some questions:

1. How was the cohort selected? Have these individuals had exome sequencing and/or are they negative for variants in genes associated with BAV (e.g. NOTCH1, GATA5, ROBO4)? Unless I missed it, I didn't see mention of this in the manuscript or materials and methods.

2. It would be informative to randomly select a control group of individuals who are negative for BAV and assess the frequency of these variants. I am nearly certain these variants are present in the control group, but at what frequency? That could potentially help bolster the argument that poly-His variants are much more commonly represented in BAV, rather than stating "reported frequencies in gnomAD for these insertions and deletions are in accordance with BAV frequency in the general population."

3. If these variants are present at a similar frequency in the BAV-negative control population, it would suggest the variants are not acting alone to cause the phenotypes, but may contribute to the overall relative risk when factoring in polygenic risk scores. Genetic findings to date would suggest the majority of BAV is polygenic, similar to many vascular disorders.

The authors present fairly convincing evidence that the Hoxa1 variants decrease protein function and are acting in a dominant negative fashion in cells, and are sufficient to affect valve development in zebrafish. They provide convincing data in Hoxa1 knockout mice showing that Hoxa1 is necessary for neural crest migration and patterning of the heart and aortic valves.

Ideally, it would have been interesting to generate a genetic mouse model, in which an insertion or deletion was engineered in the poly-His tract. As such, there is a bit of a disconnect between the zebrafish and mouse models.

- For example, are the observed phenotypes in zebrafish due to faulty neural crest cell migration as well? This may very well be the case, but it wasn't shown.
- Would some of the same phenotypes and gene expression changes be found in the Poly-His mutant mice versus the constitutive null? This would add strong translational and clinical relevance, and may point to other gene candidates.
- If we assume the poly-his variant causes a dominant negative effect, and decreases protein function to less than 50%, do the humans show any additional phenotypes?
 - o Is there evidence for internal carotid artery malformations or other changes to the great arteries?
 - o Are there any oculomotility findings, sensorineural hearing loss, etc as found in BSAS or ABDS?

Minor Points related to Figures:

1. Figure 2: it would be helpful to include another representative protein blot
2. Figure 4:

- a. Label "C" is missing in figure panel
- b. Panel D, yellow appears green in the image
- c. Asterisks should be explained in panels B and C
- d. Panel G, change E18.5 text from white to black
- e. In A, 11 total mice are listed in the mutant, but only 6 are listed under phenotype

3. Figure 5:

- a. Label PECAM on image A
- b. It's hard to see which exact areas are being counted. Outlining the areas of interest with hash marks would be helpful.

Overall, this study has strong merit but can be improved by addressing the questions above. I understand the perils associated with generating mouse models engineered to express single amino acid variants (I myself have experienced them first hand). However, doing so could have yielded some valuable insights, especially because I am still somewhat skeptical about whether the poly-his variants are sufficient alone, or to what extent they contribute to the overall risk of BAV in humans. I think the authors need to think and comment more about this in the results and discussion, and discuss more about gene penetrance and whether or not these variants are acting alone or in combination with other factors that predispose to BAV. As such, the main weakness of this study is assessing the relative impact of the "pathogenic" variants and how tightly associated these variants are with BAV in humans.

The bulk of the data in this paper mitigates these concerns to some degree, and the phenotype also has reduced penetrance in *Hoxa1* null animals, suggesting aortic valve phenotypes would be rare in mice engineered to express poly-His variants (and also suggesting these variants are not acting alone, but rather increase the relative risk in humans). I think the study has some important clinical merits (and provides important fundamental biological insights) and will lead to additional studies that shed further light on the functional relationship of *Hoxa1* poly His-variants to human BAV.

Reviewer #2 (Remarks to the Author):

Odelin et al report on functional studies of HOXA1 sequence variants that they discovered in a cohort of 333 human BAV samples. Experiments to demonstrate the role of HOXA1 in valve development include knockdown of the zebrafish orthologs *hoxa1a* and *hoxb1b* with rescue by human HOXA1 and analysis of *hoxa1*^{-/-} mice. They conclude that HOXA1 is required for aortic valve development because it is essential for the normal migration of neural crest cells. The role of HOXA1 in heart development was previously known based on the phenotypes of human patients with truncating mutations in HOXA1. The novel aspect of this work is the claim that relatively common variation in the length of the N-terminal polyhistidine repeat tract of HOXA1 causes BAV in the apparent absence of other significant cardiovascular anatomic defects. There are several significant methodologic flaws that appear to undermine this conclusion:

1. The polyhistidine repeat tract in question is not evolutionarily conserved in length, but varies quite widely between species.
2. The variants that they describe are quite common in the general population. They did not perform a case-control analysis to confirm that these variants are enriched in BAV cases.
3. The phenotype of *Hoxa1*^{-/-} mice or zebrafish probably does not resemble the expected phenotype of heterozygous *Hoxa1* polyhistidine length mutants, which would be expected to produce some protein, even if the half-life is decreased. Most of the conclusions about the phenotype of BAV caused by *hoxa1* mutation are based on analysis of the mice.
4. What are the valve phenotypes of human patients with HOXA1 polyhistidine variants? The authors observed a distinctive reduction of the non-coronary cusp in the mice. Therefore, it is striking that they omitted this information.

5. HOXA1 is required for the development of many pharyngeal arch structures, which are all affected in patients with HOXA1 null mutations. If the mutations that they describe truly decrease HOXA1 function, why do they not cause more extensive and severe phenotypes than isolated BAV and thoracic aortic aneurysms?

6. To rescue the zebrafish morphants, they injected the wild-type human HOXA1 mRNA. Why do they not report the injection of the HOXA1 mutant mRNAs into the morphants? Were the polyhistidine mutants able to rescue the morphant phenotypes? It is unclear why they only chose to inject the mutant mRNAs into wild-type embryos. The comparison between the two experiments would demonstrate that the mutation has a dominant negative effect, as they assert. I find no convincing evidence of this in the data that they provided.

The *hoxa1* mouse data was the most methodologically sound and convincing part of the manuscript. The human and zebrafish data do not meet this standard. In my opinion, these data do not support the authors' claims about the relevance of HOXA1 to human BAV.

Point-by-point reply to the reviewer's comments for the manuscript "Variations in the poly-histidine repeat motif of HOXA1 contribute to bicuspid aortic valve"

First, we want to thank all reviewers for their thoughtful comments and suggestions which helped us to significantly improve the manuscript. In the revised version of the manuscript, we have addressed all issues raised as outlined in the point-by-point reply below.

REVIEWER COMMENTS

Reviewer #1 (Remarks to the author):

The authors present data in humans, zebrafish, and mouse indicating that insertions or deletions in the poly-his tract of HOXA1 predispose to BAV. HOXA1 is essential for vascular development in humans, and this study adds further evidence for a role in human aortic valve development. The paper is generally well written, and the figures are clearly presented. As such, this study provides important biological and clinical insights into the functions of Hoxa1/HOXA1 for aortic valve development.

We thank the reviewer for highlighting the relevance of our data and suggesting additional analyses which helped us to improve our manuscript significantly.

As the authors point out, insertions or deletions in the HOXA1 poly-his tract are relatively common in humans. In the early 2000s, for example, variants in this region were investigated for association with autism. This raises some questions:

1. How was the cohort selected? Have these individuals had exome sequencing and/or are they negative for variants in genes associated with BAV (e.g. NOTCH1, GATA5, ROBO4)? Unless I missed it, I didn't see mention of this in the manuscript or materials and methods.

We agree with the reviewer that the first studies exploring HOXA1 poly-His tract were related to autism. These studies identified deletion or insertion in the HOXA1 poly-His tract in patients with autism and demonstrated that longer histidine repeats result in accelerated apoptosis (Paraguisson et al., 2005). This identification followed the first identification of a homozygous truncating mutation in *HOXA1* in patients with facial weakness, deafness, malformations of the cerebral vasculature, and autism spectrum disorder (Tischfield et al., 2005). In our study patients were prospectively included in the cohort after evaluation of their clinical presentations and echocardiography with BAV associated with leaking and stenosis degeneration. Patients with another clinical disorders such as autism were excluded (Theron et al., 2021). To further explore the presence of variants BAV-related genes, we performed whole exome sequencing (WES) on BAV patients who carried insertion or deletion in the poly-His tract of HOXA1. Exome data were analyzed with a primary focus on BAV genes including *NOTCH1*, *GATA4*, *ROBO4*, *MUC4*, and *GATA5* but also on genes implicated in valvular and aortic defects. The only relevant variants found were *NOTCH1*: p.(Pro649Thr), *NOTCH1*: p.(Thr123Met) and *GATA5*: p.(Leu233Pro). The variants in *NOTCH1* gene were either benign or with conflicting interpretation. Interestingly, the *GATA5* variant was previously identified in an individual with BAV, but no functional evidence was reported to support its implication in

BAV. We believe that these findings together with the frequency of deletion HOXA1 can explain the variable expressivity observed in some patients with BAV.

We have included these new results on page 6 of the revised version of the manuscript and stated that insertion or deletion in the poly-His tract of HOXA1 is not often frequently associated with variants in BAV genes.

“To further examine if variants in genes associated with BAV (e.g. NOTCH1, GATA4, GATA5, MUC4, and ROBO4) and implicated in valvular and aortic defects were present in individuals carrying deletion or insertion in the poly-histidine tract of HOXA1, we performed exome sequencing^{16,17,19,24}. After exome sequencing analysis no relevant variants were identified except for three missense variants in NOTCH1 and GATA5 genes. One patient with deletion of 1 histidine (-1His) carried a heterozygous NOTCH1 variant (p.Pro649Thr). The minor allele frequency (MAF) of this variant (rs780710009) is 0.00001 in gnomAD (Supplementary Table 3). This variant is located in an EGF-like repeat domain of NOTCH1 with a conflicting interpretation of pathogenicity. Another BAV patient with aneurysm was carrying a heterozygous NOTCH1 variant (p.Thr123Met), which is considered as a polymorphism (Supplementary Table 3), and a heterozygous variant in the GATA5 gene (p.Leu233Pro), which has already been identified in one individual with BAV but without functional evidence¹⁷. Thus, exome analysis showed that variants in BAV genes were not frequently found in individuals with deletion or insertion in the poly-histidine tract of HOXA1 (2/29).”

2. It would be informative to randomly select a control group of individuals who are negative for BAV and assess the frequency of these variants. I am nearly certain these variants are present in the control group, but at what frequency? That could potentially help bolster the argument that poly-His variants are much more commonly represented in BAV, rather than stating “reported frequencies in gnomAD for these insertions and deletions are in accordance with BAV frequency in the general population.”

Following the reviewer’s suggestion, we used a control cohort composed of 856 individuals from France (The FranceGenRef panel) to assess the frequency of the variants identified in our cohort of BAV. We found that +1His, +1His^{Arg} and -3His^{Arg} variants were more frequently identified in our cohort than in these controls (0.60 vs. 0.18; 0.15 vs. 0; 0.45 vs. 0.12; see Table 2 and Supplementary Table 2).

We have included these new results on page 5-6 of the revised manuscript and stated that most variants identified in HOXA1 are more commonly observed in BAV patients than in a control cohort.

“To assess the frequency of these variants, we used a control cohort composed of 856 individuals from France (The FranceGenRef panel; see materials and methods). Using this cohort, we found that except for the -1His variant, the +1His, +1His^{Arg} and -3His^{Arg} variants are more commonly represented in BAV patients than in controls (Table 2; Supplementary Table 2). Taken together, our data suggest that variants in the poly-histidine tract of HOXA1 are potentially associated with BAV.”

3. If these variants are present at a similar frequency in the BAV-negative control population, it would suggest the variants are not acting alone to cause the phenotypes, but may contribute to the overall relative risk when factoring in polygenic risk scores. Genetic findings to date would suggest the majority of BAV is polygenic, similar to many vascular disorders.

We agree with the reviewer that the majority of BAVs are polygenic and our results are consistent with this. In the revised discussion we now include a paragraph saying that we cannot exclude that *HOXA1* variants act in concert with other genetic factors to finally contribute to the onset of the BAV phenotype.

The authors present fairly convincing evidence that the *Hoxa1* variants decrease protein function and are acting in a dominant negative fashion in cells, and are sufficient to affect valve development in zebrafish. They provide convincing data in *Hoxa1* knockout mice showing that *Hoxa1* is necessary for neural crest migration and patterning of the heart and aortic valves.

Ideally, it would have been interesting to generate a genetic mouse model, in which an insertion or deletion was engineered in the poly-His tract. As such, there is a bit of a disconnect between the zebrafish and mouse models.

- For example, are the observed phenotypes in zebrafish due to faulty neural crest cell migration as well? This may very well be the case, but it wasn't shown.

As the reviewer points out, the neural crest migration is defective in the *Hoxa1-null* mutant mice. To further examine whether neural crest cell migration is also disrupted in zebrafish lacking *hoxa1a* we performed additional experiment using transgenic *sox10:eGFP* line in *hoxa1a* morphants and crispants. As illustrated in the novel supplementary Fig. 11 and Fig. 12, decrease or absence of *hoxa1a* in zebrafish results in abnormal *sox10* patterning.

We have included these new results on page 12 of the revised manuscript and stated that similar to the mouse, *hoxa1a* could be involved in neural crest cell migration.

“To determine whether neural crest migration was also affected in zebrafish, we examined hoxa1a morphants and crispants in a Tg(sox10:eGFP) line. At 24 or 48hpf, we did not detect any obvious difference between injected and non-injected fish (Supplementary Fig. 11). However, at 3dpf, we observed a GFP signal in the pharyngeal arch and OFT region which was absent from the morphant and crispant specimens (Supplementary Fig. 12a-c). We next sought to investigate whether overexpression of the variants could also affect sox10 patterning. Injection of the WT or +1His variant mRNA did not seem to affect GFP expression (Supplementary Fig. 12d,e). However, both -1His or -3His^{Arg} induced a loss of GFP expression in this area (Supplementary Fig. 12f,g) suggesting that, similar to the mouse, hoxa1a could be involved in neural crest cell migration.”

- Would some of the same phenotypes and gene expression changes be found in the Poly-His mutant mice versus the constitutive null? This would add strong translational and clinical relevance, and may point to other gene candidates.

We thank the reviewer for this helpful suggestion. Using CRISPR-Cas9 technology we generated a novel *Hoxa1*^{-1His} knock-in mouse line and found that like the knock-out, CRISPR-

Cas9 mice presented BAV phenotype with incomplete penetrance. Furthermore, we revisited the phenotype of *Hoxa1*^{WM-AA} knock-in mice, which amino acid substitutions in the hexapeptide motif of Hoxa1 that abolish its interaction with its co-factor Pbx and its activity on known target enhancers in cellular models. Similar to *Hoxa1*⁻ knock-out and *Hoxa1*^{-1His} knock-in embryos, we observed BAV in *Hoxa1*^{WM-AA} knock-in embryos, suggesting that reduced Hoxa1 activity results in abnormal aortic valve development.

We have included these new results on pages 9-10 of the revised manuscript and stated that *HOXA1* mutations are not the leading cause of the BAV phenotype but participate in it.

“We next sought to evaluate the in vivo consequence of the poly-histidine tract length modifications of HOXA1 observed in BAV patients by establishing CRISPR-Cas9 alleles in mouse (see Methods). Like Hoxa1 knock-out, homozygous Hoxa1^{-1His} mice exhibited a BAV phenotype with incomplete penetrance. At E18.5, 17% (4/23) of homozygous Hoxa1^{-1His} embryos had BAV (Fig. 4i,j; Table 4). Of note, a BAV phenotype was never observed in Hoxa1^{+/+} littermate embryos (n=21). Interestingly, 9% (5/52) of BAV was also observed in homozygous Hoxa1^{WM-AA} knock-in embryos (Supplementary Fig. 7; Table 4), which carry amino acid substitutions in the hexapeptide motif of Hoxa1 that abolish its interaction with its co-factor Pbx and its activity on known target enhancers in cellular models³⁹. This observation suggests that Hoxa1 activity is required for aortic valve development. The incomplete penetrance of the BAV phenotype observed in Hoxa1 knock-out and knock-in mice is in line with several mouse models of valve disease, which usually show incomplete penetrance, such as Nkx2-5 (8.2% for BAV), Gata5 (25% for BAV), Krox20 (30% for BAV), and Robo4 (17.9% in male)^{12,19,40,41}. Collectively, these findings indicate that reduction of Hoxa1 activity contributes to a BAV phenotype.”

- If we assume the poly-his variant causes a dominant negative effect, and decreases protein function to less than 50%, do the humans show any additional phenotypes?

- o Is there evidence for internal carotid artery malformations or other changes to the great arteries?

- o Are there any oculomotility findings, sensorineural hearing loss, etc as found in BSAS or ABDS?

As the reviewer points out, the poly-histidine variants cause a dominant negative effect and decreases protein function to less than 50%. As suggested by the reviewer we verified whether patients carrying these variants have additional malformations. We found that only 3 patients have artery malformations, and none of them have ocular or ear defects. These observations suggest that the poly-histidine variants identified in BAV patients have different effects from the variants identified in BSAS or ABDS patients.

Minor Points related to Figures:

1. Figure 2: it would be helpful to include another representative protein blot

As requested, Figure 2 now includes another representative protein blots.

2. Figure 4:

- a. Label “C” is missing in figure panel

Label “C” is now included.

- b. Panel D, yellow appears green in the image

Color is modified.

c. Asterisks should be explained in panels B and C

Asterisks are now explained in the figure legend.

d. Panel G, change E18.5 text from white to black

E18.5 text is now changed.

e. In A, 11 total mice are listed in the mutant, but only 6 are listed under phenotype

We apologize for the confusing presentation of data. The number of BAV phenotype is now listed in Table 4.

3. Figure 5:

a. Label PECAM on image A

Label is now included.

b. It's hard to see which exact areas are being counted. Outlining the areas of interest with hash marks would be helpful.

We apologize for it. Areas of interest are now outlined.

Reviewer #2 (Remarks to the author):

Odelin et al report on functional studies of HOXA1 sequence variants that they discovered in a cohort of 333 human BAV samples. Experiments to demonstrate the role of HOXA1 in valve development include knockdown of the zebrafish orthologs *hoxa1a* and *hoxb1b* with rescue by human HOXA1 and analysis of *hoxa1*^{-/-} mice. They conclude that HOXA1 is required for aortic valve development because it is essential for the normal migration of neural crest cells. The role of HOXA1 in heart development was previously known based on the phenotypes of human patients with truncating mutations in HOXA1. The novel aspect of this work is the claim that relatively common variation in the length of the N-terminal polyhistidine repeat tract of HOXA1 causes BAV in the apparent absence of other significant cardiovascular anatomic defects. There are several significant methodologic flaws that appear to undermine this conclusion:

We thank the reviewer for acknowledging the relevance of our work and for pointing our attention to the need for improving the justification of our conclusion, which helped us to significantly enhance this aspect of our work.

1. The polyhistidine repeat tract in question is not evolutionarily conserved in length, but varies quite widely between species.

We agree with the reviewer that the length of the poly-histidine repeat varied during evolution. Following this comment, we performed a further analysis of the conservation of HOXA1 amino-acid sequences and found that the poly-histidine tract is absent from the bony fish that have undergone genome duplication but is present in the cartilaginous fish (*Heterodontus* and *Callorhinchus*). Such conservation demonstrates the conservation of this domain in common ancestral and the role plays by this poly-histidine tract domain in HOXA1 activity. We believe the novel alignment is more informative and can better clarify the conservation of this domain during evolution. We have included this in the revised supplementary Figure 1 and discuss these findings in the discussion part of the manuscript starting on page 14.

2. The variants that they describe are quite common in the general population. They did not perform a case-control analysis to confirm that these variants are enriched in BAV cases.

Following the reviewer's suggestion, we used a control cohort composed of 856 individuals from France (The FranceGenRef panel) to assess the frequency of the variants identified in our cohort of BAV. We found that +1His, +1His^{Arg} and -3His^{Arg} variants were more frequently identified in our cohort than in these controls (0.60 vs. 0.18; 0.15 vs. 0; 0.45 vs. 0.12; see Table 2 and Supplementary Table 2).

We have included these new results on page 5-6 of the revised manuscript and stated that most variants identified in HOXA1 are more commonly observed in BAV patients than in a control cohort.

"To assess the frequency of these variants, we used a control cohort composed of 856 individuals from France (The FranceGenRef panel; see materials and methods). Using this cohort, we found that except for the -1His variant, the +1His, +1His^{Arg} and -3His^{Arg} variants are more commonly represented in BAV patients than in controls (Table 2; Supplementary Table 2). Taken together, our data suggest that variants in the poly-histidine tract of HOXA1 are potentially associated with BAV."

3. The phenotype of Hoxa1 ^{-/-} mice or zebrafish probably does not resemble the expected phenotype of heterozygous Hoxa1 polyhistidine length mutants, which would be expected to produce some protein, even if the half-life is decreased. Most of the conclusions about the phenotype of BAV caused by hoxa1 mutation are based on analysis of the mice.

As the reviewer points out correctly, the phenotype of *Hoxa1*⁻ knock-out mice is associated with a complete loss of *Hoxa1* function, whereas BAV patients carry heterozygous variants in *HOXA1*. However, our study demonstrates that length variations of the poly-histidine tract result to a dominant negative effect as shown by the luciferase assay experiment and injections in zebrafish. Importantly, our revised version includes new findings showing that reduction of the poly-histidine tract in the mouse results also to BAV phenotype (these new results are included on pages 9-10). Furthermore, we revisited the phenotype of *Hoxa1*^{WM-AA} knock-in mice, which amino acid substitutions in the hexapeptide motif of Hoxa1 that abolish its interaction with its co-factor Pbx and its activity on known target enhancers in cellular models. Like *Hoxa1*⁻ and *Hoxa1*^{-1His} mice, we also observed BAV phenotype in *Hoxa1*^{WM-AA} knock-in embryos (Supplementary Figure 7). Thus, these findings support our conclusion that reduction of HOXA1 activity contributes to BAV phenotype. Of course, we cannot exclude that the *HOXA1* variants act in concert with other genetic factors to contribute to the onset of the BAV phenotype as stated in the discussion. This point is now more clearly discussed in the revised version of our manuscript.

4. What are the valve phenotypes of human patients with HOXA1 polyhistidine variants? The authors observed a distinctive reduction of the non-coronary cusp in the mice. Therefore, it is striking that they omitted this information.

We agree with the reviewer that it would be interesting to compare the valve phenotype of human patients with variants in the poly-histidine tract of HOXA1 to the phenotype that we described in the *Hoxa1* mutant mouse model. However, abnormal growth of non-coronary leaflet, as observed in the mouse model, will be difficult to detect in humans by regular echocardiographic clinical examination. Therefore, it is why we conclude that “a classification that could recognize all types of BAV would be useful to better identify the genetic causes of this pathology”. Therefore, we extended our discussion starting page 17.

To further verify the valvular phenotype in BAV patients with variants in the poly-histidine tract of HOXA1, we compared the frequency of different types of BAV to those described in the general population (Sievers & Schmidtke 2007). Although the frequency of BAV type 1 is similar to what we observed, BAV type 0 is more frequently represented in our cohort than in the general population (20% vs. 7%). This observation may reflect the contribution of HOXA1 in the phenotype. We have included the new results on page 5.

“The clinical BAV phenotype of these patients can be found in supplementary Table 1. Interestingly, BAV type 0 is more frequently observed in patients with variant in the poly-histidine tract of HOXA1 than in the general population (20% vs. 7%; p=0.03)⁸.”

5. HOXA1 is required for the development of many pharyngeal arch structures, which are all affected in patients with HOXA1 null mutations. If the mutations that they describe truly decrease HOXA1 function, why do they not cause more extensive and severe phenotypes than isolated BAV and thoracic aortic aneurysms?

As the reviewer points out, our lab as other has shown that *Hoxa1* is required for the development of pharyngeal arch structures. Our current work demonstrates that the poly-histidine variants cause a dominant negative effect and reduces *Hoxa1* protein activity to less than 50%. We do not claim that these variants correspond to null mutations. As suggested by the reviewer, we verified whether patients carrying these variants have additional malformations. We found that only 3 patients have artery malformations, and none of them have ocular or ear defects. Thus, the poly-histidine variants might affect the developmental processes which are the most sensitive to a temporal (half-life decrease) loss-of-function of the protein. Furthermore, our observations suggest that the poly-histidine variants identified in BAV patients have different effects from the variants identified in patients with Bosley Salih Alorainy syndrome which are essentially truncating mutations (Tischfield et al. 2005; Bosley et al. 2008).

6. To rescue the zebrafish morphants, they injected the wild-type human HOXA1 mRNA. Why do they not report the injection of the HOXA1 mutant mRNAs into the morphants? Were the polyhistidine mutants able to rescue the morphant phenotypes? It is unclear why they only chose to inject the mutant mRNAs into wild-type embryos. The comparison between the two experiments would demonstrate that the mutation has a dominant negative effect, as they assert. I find no convincing evidence of this in the data that they provided.

We agree that injecting the human mutated mRNA into the morphants would give us functional information about the mutated proteins. In fact, we have performed these experiments but co-injecting both morpholinos and variants gave a very strong phenotype as

a result of the effects of both components. Therefore, we decided not to include these results since injecting the mutated mRNA alone was giving us information about the dominant effect of the variants. Nevertheless, if the reviewer thinks we should include this information, we can add the data in the manuscript.

The *hoxa1* mouse data was the most methodologically sound and convincing part of the manuscript. The human and zebrafish data do not meet this standard. In my opinion, these data do not support the authors' claims about the relevance of HOXA1 to human BAV.

We are sorry to see that our results and conclusion did not convince the reviewer. We hope that our additional experiments and new results (CRISPR-Cas9 mouse line, novel zebrafish data...) and novel important finding with the *Hoxa1*^{-1^{His}} mice will succeed in convincing the reviewer of the involvement of the relevance of HOXA1 in human BAV.

REVIEWERS' COMMENTS

Reviewer #1 (Remarks to the Author):

The inclusion of additional data, including analyzing the frequency of HIS variants in unaffected controls and phenotypic analysis of a -1HIS knock-in model, have significantly improved the overall conclusions of the study. Although BAV was present in 17% of homozygous animals (and no mention of phenotypes in heterozygote animals, which would be more consistent with a dominant negative effect) the presence of BAV nonetheless strongly suggests that the -1HIS variants exerts a loss-of-function effect on the protein. This agrees with findings in KO HOXA1 mice and supports the pathogenic nature of the HIS variants in humans.

Inclusion of additional text, including describing the polygenic nature of BAV, is more consistent with the overall findings and the frequency of these HIS variants in the general population (although at lower frequency). In my opinion, it is plausible that HIS variants, likely in combination with other genetic factors, predispose to BAV.

Also, if BAV was not analyzed in heterozygous -1HIS knock-in mice, that should be mentioned in the text. If it was analyzed and BAV was not present in heterozygotes, that should also be noted.

Overall, the combination of human/mouse/zebrafish data support HOXA1 HIS variants as contributing to the cause of BAV in humans.

Reviewer #2 (Remarks to the Author):

The authors did not satisfactorily add to the evidence that HOXA1 variants genetically contribute to BAV in humans:

1. The polyhistidine tract is not evolutionarily conserved.
2. The frequencies of the polyhistidine variants are not enriched in BAV cases.
3. No family data is presented to show that HOXA1 variants segregate with BAV.
4. The evidence for a dominant negative phenotypic effect is unconvincing because it is based on co-injection of mutant and wild-type HOXA1 in a luciferase reporter assay. Is it possible that transactivation by the mutant protein is weaker than wild-type? If the mutant protein is degraded, as they assert, then why would it affect wild-type function?
5. HOXA1 human mutations affect multiple systems and cause severe developmental phenotypes. If HOXA1 contributes to BAV disease, were these other phenotypes present?

Due to these significant limitations, this manuscript functions best as a developmental analysis of BAV in animal models and is less relevant to human disease. In this sense, their manuscript is similar to other publications showing that many mutations that cause BAV in mice do not translate to human BAV..

Point-by-point reply to the reviewer's comments for the manuscript "Variations in the poly-histidine repeat motif of HOXA1 contribute to bicuspid aortic valve in mouse and zebrafish"

First, we want to thank all reviewers for their positive comments on our manuscript. In the final revised version of the manuscript, we have addressed all issues raised as outlined in the point-by-point reply below.

REVIEWER COMMENTS

Reviewer #1 (Remarks to the author):

The inclusion of additional data, including analyzing the frequency of HIS variants in unaffected controls and phenotypic analysis of a -1HIS knock-in model, have significantly improved the overall conclusions of the study. Although BAV was present in 17% of homozygous animals (and no mention of phenotypes in heterozygote animals, which would be more consistent with a dominant negative effect) the presence of BAV nonetheless strongly suggests that the -1HIS variants exerts a loss-of-function effect on the protein. This agrees with findings in KO HOXA1 mice and supports the pathogenic nature of the HIS variants in humans.

Inclusion of additional text, including describing the polygenic nature of BAV, is more consistent with the overall findings and the frequency of these HIS variants in the general population (although at lower frequency). In my opinion, it is plausible that HIS variants, likely in combination with other genetic factors, predispose to BAV.

We thank the reviewer for his/her comments on our revised manuscript.

Also, if BAV was not analyzed in heterozygous -1HIS knock-in mice, that should be mentioned in the text. If it was analyzed and BAV was not present in heterozygotes, that should also be noted.

We agree with the reviewer that this result is important. This result is provided in the revised version of our manuscript (line 252; page 9).

Reviewer #2 (Remarks to the author):

The authors did not satisfactorily add to the evidence that HOXA1 variants genetically contribute to BAV in humans:

We are sorry to read that our new results including CRISPR-Cas9 mouse line, and novel zebrafish data did not convince the reviewer.

1. The polyhistidine tract is not evolutionarily conserved.

Although the polyhistidine tract is not conserved in zebrafish we have shown that is conserved in the cartilaginous fish (*Heterodontus* and *Callorhinchus*). Such conservation demonstrates the conservation of this domain in common ancestral.

While mouse *Hoxa1* has a repetition of 11 His, which is different from the 10 His in Human, we demonstrated that deletion of one histidine in the poly-histidine tract in mice leads to a BAV phenotype. This result supports the important role of the polyhistidine tract in evolutionarily distant species.

2. The frequencies of the polyhistidine variants are not enriched in BAV cases.

Following the reviewer's previous suggestion, we used a control cohort composed of 856 individuals from France (The FranceGenRef panel) to assess the frequency of the variants identified in our cohort of BAV. We found that +1His, +1His^{Arg} and -3His^{Arg} variants were more frequently identified in our cohort than in the control population used (0.60 vs. 0.18; 0.15 vs. 0; 0.45 vs. 0.12; see Table 2 and Supplementary Table 2). Therefore, the frequency is slightly increased in BAV cases compared to the control cohort.

3. No family data is presented to show that HOXA1 variants segregate with BAV.

As indicated in our study we used a local cohort of patients with BAV to sequence *HOXA1* gene. Unfortunately, we did not have access to the related family of these BAV patients to perform a co-segregation analysis. However, BAV is known to have an autosomal dominant pattern of inheritance with reduced penetrance and variable expressivity (PMID: 22701807). Therefore, allelic segregation cannot be used to exclude causative gene in BAV.

Our revised manuscript include discussion on the limitations of our study.

4. The evidence for a dominant negative phenotypic effect is unconvincing because it is based on co-injection of mutant and wild-type HOXA1 in a luciferase reporter assay. Is it possible that transactivation by the mutant protein is weaker than wild-type? If the mutant protein is degraded, as they assert, then why would it affect wild-type function?

The dominant negative phenotypic effect was not only based on co-injection of mutant and wild-type HOXA1 in a luciferase assay since we also evaluated whether the expression of the dominant negative human *HOXA1* variants in the zebrafish had an effect. Expressing any of the dominant negative variants (+1His, -1His or -3His^{Arg}) also had a significant impact on aortic valve development when compared to wild-type *HOXA1* expression (Fig. 3n).

5. HOXA1 human mutations affect multiple systems and cause severe developmental phenotypes. If HOXA1 contributes to BAV disease, were these other phenotypes present?

As indicated in our revised manuscript some BAV patients had other phenotypes (Supplementary Table 3). Interestingly, homozygous *Hoxa1*^{-/-} and *Hoxa1*^{WM-AA} mice presented also great artery defects. These observation are now included in Supplementary Table 4.